# Relative performance of empirical and physical models in assessing the seasonal and annual glacier surface mass balance of Saint Sorlin glacier (French Alps)

Marion REVEILLET[1][*], Delphine SIX[1], Christian VINCENT[1], Antoine RABATEL[1], Marie DUMONT[2], Matthieu LAFAYSSE[2], Samuel MORIN[2], Vincent VIONNET[2] and Maxime LITT[3-1].

[1] Univ. Grenoble Alpes, CNRS, IRD, Institut des Géosciences de l'Environnement (IGE, UMR 5001), F-38000 Grenoble, France

[2] Météo-France CNRS, CNRM/CEN UMR 3589, Météo-France, CNRS, Grenoble, France

[3] ICIMOD, GPO Box 3226, Kathmandu, Nepal

[*] now at: Centro de Estudios Avanzados en Zonas Áridas (CEAZA), ULS-Campus Andrés Bello, Raúl Britan 1305, La Serena, Chile

*Correspondence to*: Marion REVEILLET (marion.reveillet@ceaza.cl)

**Abstract.** This study focuses on simulations of the seasonal and annual surface mass balance (SMB) of Saint-Sorlin Glacier (French Alps) for the period 1996-2015 using the detailed SURFEX/ISBA-Crocus snowpack model. The model is forced by SAFRAN meteorological reanalysis data, adjusted with AWS measurements to ensure that simulations of all the energy balance components, in particular turbulent fluxes, are accurately represented with respect to the measured energy balance. Results indicate good model performance for the simulation of summer SMB when using meteorological forcing adjusted with in-situ measurements. Model performance however strongly decreases without in-situ meteorological measurements. The sensitivity of the model to meteorological forcing indicates a strong sensitivity to wind speed, higher than the sensitivity to ice albedo. Compared to an empirical approach, the model exhibited better performance for simulations of snow and firn melting in the accumulation area and similar performance in the ablation area when forced with meteorological data adjusted with nearby AWS measurements. When such measurements were not available close to the glacier, the empirical model performed better. Our results suggest that simulations of the evolution of future mass balance using an energy balance model requires very accurate meteorological data which are not reliable from the climatic scenarios. Given the uncertainties in the temporal evolution of the relevant meteorological variables and glacier surface properties in the future, empirical approaches based on temperature and precipitation could be more appropriate for simulations of glaciers in the future.

## 1. Introduction

The surface mass balance (SMB) of mountain glaciers is sensitive to climate change and contributes to the hydrological regime of high alpine catchments (IPCC, 2013). Understanding the physical processes that link local meteorology to glacier melt is necessary to properly simulate changes in glacier SMB in the context of global warming.

Several studies have successfully used various calibrated temperature-index models (TIM) to simulate glacier melt response to meteorological forcing (Braithwaite and Olesen, 1989; Hock, 2003; Pellicciotti *et al.,* 2005). These approaches can be used over short time periods (typically a few years), but the relevance of the calibrated parameters over longer time periods is difficult to assess for several reasons including: (i) the lack of long term *in-situ* meteorological measurements available close to the study site, (ii) the temporal variations of melt sensitivity to temperature and (iii) the fact that the physical link between temperature and melt is not direct (Huss *et al.,* 2009; Gabbi *et al.,* 2014; Réveillet *et al.*, 2017). In addition, transferring parameters determined for an instrumented glacier to another site decreases model performance (Carenzo *et al.,* 2009; Réveillet *et al.,* 2017).

On the other hand, physical approaches consider all energy exchanges between the glacier and the atmosphere and are able to represent snow melt spatial and temporal variability, such as those related to albedo variations that are hard to represent in TIM models. Such approaches offer higher transferability over time (e.g., MacDougall and Flowers, 2011) but require more accurate meteorological forcing (e.g., Gabbi *et al.,* 2014). Many energy balance studies have been performed to assess surface-atmosphere interactions over ice or snow surfaces based on automatic weather stations (AWS) deployed on glaciers (e.g., Oerlemans and Klok, 2002; Sicart *et al.,* 2008; Senese *et al.,* 2012; Cullen and Conway, 2015). Physically based models perform well for SMB simulations when AWS measurements are available on the study site (e.g., Six *et al., 2009)* and enable a quantification of each component of the energy budget and their impact on melting. However, due to the need for accurate meteorological data and the difficulty of maintaining AWSs on glaciers, this approach is generally used over short time periods (typically a few months), except for a few studies based on permanent AWSs set up on glaciers (e.g., Oerlemans *et al.,* 2009; Sicart *et al.,* 2011).

These physical models, using *in-situ* meteorological data or coupled with atmospheric models (e.g., Lefebre, 2003; Mölg and Kaser, 2011) or forced by meteorological reanalysis (e.g., Gerbaux *et al.,* 2005), provide an opportunity to determine the spatial distribution of SMB evolution over longer periods. The simulation of seasonal SMB changes requires accurate modelling of energy exchanges over both ice and snow surfaces. Detailed snowpack models such as Crocus (Brun *et al.,* 1989), SNOWPACK (Lehning *et al.,* 1999) or Snow-SVAT (Tribbeck *et al.,* 2004) have been developed and some have been applied to glaciers (e.g., Obleitner and Lehning, 2004; Gerbaux *et al.*, 2005; Dumont *et al.*, 2012; Lejeune *et al.*, 2013; Sauter and Obleitner, 2015). Due to the lack of measurements and the complexity of measuring each of the components of the energy balance (especially turbulent fluxes), physically based models are generally calibrated by adjusting certain parameters (e.g., roughness length to quantify turbulent fluxes) to fit with SMB measurements (e.g. Dumont *et al.,* 2012).

The goal of our study is to evaluate the performance of a physical model in simulating seasonal SMB and to compare its

performance and the associated uncertainties to those obtained with a temperature-index model in order to determine the most appropriate approach for SMB simulations, especially for projections over long time periods. In the Alps, the temporal variability of the annual SMB is mainly driven by summer SMB variability (e.g., Six and Vincent, 2014). For this reason, many studies have focused on ablation modelling. However, simulated summer SMB and associated uncertainties strongly depend on the winter SMB (Réveillet *et al.,* 2017), highlighting the need for a quantification of the sensitivity of annual SMB to both seasonal components.

For these purposes, we use the detailed SURFEX/ISBA-Crocus snowpack model (Vionnet *et al.,* 2012), driven by SAFRAN meteorological reanalysis data (Durand *et al.,* 2009), to simulate the SMB of Saint-Sorlin Glacier (French Alps). We first assess the accuracy of SAFRAN meteorological reanalysis data at this high elevation site using all available glaciological and meteorological measurements performed since 2005 on Saint-Sorlin Glacier. Then, the surface energy and mass balance model is calibrated using the measured energy balance to ensure that all the energy balance components are accurately represented. Next, the SMB model is evaluated using twenty years of seasonal SMB measurements (section 4.1.1) and results are compared to those obtained with temperature-index models (section 4.1.2). Section 4.1.3 focuses on annual SMB sensitivity to seasonal SMB. Finally, Crocus model sensitivity to meteorological forcing, calibration and topographic parameters is analysed in section 4.2.

## 2. Study site and data

### 2.1 Study site: Saint-Sorlin Glacier

Saint-Sorlin Glacier is located in the Grandes Rousses massif in the French Alps (Figure 1) and is monitored by the GLACIOCLIM program (https://glacioclim.osug.fr). Saint-Sorlin Glacier covers a surface area of roughly 2.5 km$^2$. The glacier flows along slopes with highly variable aspects, descending from 3460 to 2700 m a.s.l.. More details on the topographic characteristics of this glacier are provided in Six and Vincent (2014).

### 2.2 Glaciological measurements over the period 1995-2015

### 2.2.1 Seasonal surface mass balance measurements

Seasonal SMB has been monitored since 1995 using the glaciological method (Cuffey and Paterson, 2010) at about 30 measurements points (Figure 1). During summer (*i.e.* from around 15 April to 15 October, corresponding to the ablation season), the glacier is regularly visited and monthly ablation measurements are available. The uncertainties of the SMB measurements are evaluated at approximately ±0.20 m w.e. yr$^{-1}$ for winter surface mass balance (winter SMB) and ±0.15 m w.e. yr$^{-1}$ (resp. 0.30 m w.e. yr$^{-1}$) for summer surface mass balance (summer SMB) on ice (resp. snow/firn) (Thibert *et al.*, 2008). The monitoring network covers a large part of the glacier both in the accumulation and ablation areas (Figure 1). Winter SMBs are measured at each point located in the accumulation and ablation areas in late April using snow cores and

density measurements. Summer SMBs are quantified using stakes inserted in the ice/snow.

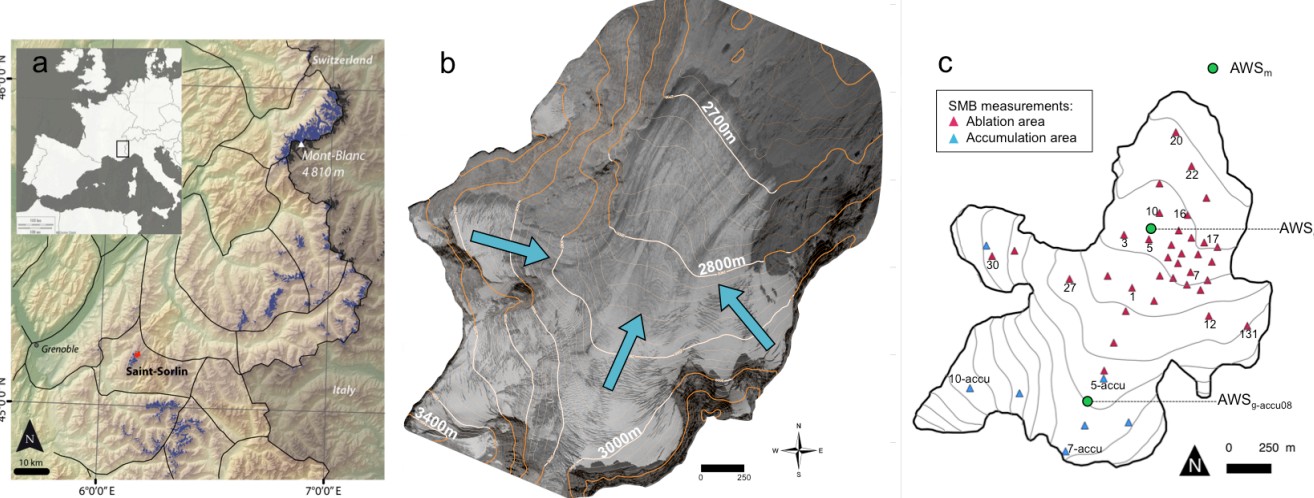

**Figure 1.** (a) Location of Saint-Sorlin Glacier in the French Alps. French glaciers are shown in blue except for Saint-Sorlin
Glacier, used for the present study, which is in red. Black lines represent SAFRAN massif outlines (adapted from Rabatel et
al., 2016). (b) Aerial photo of Saint-Sorlin glacier. Blue arrows indicate the three main glacier flow lines. (c) Map of Saint-
Sorlin Glacier with the network of in-situ SMB measurements (blue triangles in the accumulation area and red triangles in
the ablation area). Locations of automatic weather stations used in this study are represented by green circles.

### 2.2.2 Digital Elevation Models

We used three digital elevation models (DEMs) (1998, 2007 and 2014) to account for the changes in glacier geometry during
the studied period. These DEMs were derived from aerial photogrammetry and have a 10-m spatial resolution. For
consistency with the resolution of the atmospheric data described in section 2.3.3, they were, for this study, upscaled to 200-
m resolution using the kriging method (based on the default linear variogram) of SURFER mapping software (Golden
Software, LLC).

### 2.3 Meteorological data

### 2.3.1 Automatic weather stations

In the framework of GLACIOCLIM, a permanent AWS has been in operation since August 2005 on the foreland of Saint-
115 Sorlin Glacier (noted $AWS_m$ on Figure 1b). This AWS records 2-m air temperature and relative humidity (the common
sensor is housed in a mechanically aspirated shield), incoming and reflected short-wave radiation, incoming and outgoing
long-wave radiation, and wind speed and direction with a half-hour time step. $AWS_m$ data were quality checked to avoid any
problem related to a sensor malfunction: missing data were detected and reported, unrealistic values were removed and the

series was compared with series from *Meteo France* network stations in the valley to identify potential bias. A summary of the meteorological conditions at $AWS_m$ is given in the supplementary material An additional meteorological station (noted $AWS_g$ in Figure 1b) was set up in the ablation area of the glacier during each of the three summer field campaigns (2006, 2008 and 2009). It will be referred to hereafter as $AWS_{g06}$, $AWG_{g08}$ and $AWS_{g09}$ to distinguish between the different years. Note that during the 2008 field campaign, another AWS was set up in the accumulation area (noted $AWS_{g-accu08}$ in Figure 1b). Details relative to the location, the dates of records and the different sensors of these AWSs are reported in Table 1. Stations on the glaciers are mounted on masts inserted in the ice. Due to ice melt, instrument heights are not constant over time. However, at each station (except for $AWS_{g08-accu}$ where melt is limited), a sonic ranger was set up and helped determine the melt over each recorded time step. The heights of the instrument were then adjusted in our simulation using the melt determined by the sonic ranger. Every 10 to 15 days, instruments were re-adjusted manually to a set height of 2 m.

### 2.3.2 Eddy covariance system and atmospheric mast

In 2006, a summer field campaign was also conducted to measure turbulent fluxes using the Eddy covariance (EC) method (Table 1). During this campaign (9 July to 28 August 2006), an Eddy covariance system measuring the high frequency (20 Hz) wind speed components, sonic temperature and specific humidity was fixed on a mast in the ablation zone next to $AWS_{g06}$. The CSAT instrument was installed 2.00 m above the surface. The melt ranges roughly between 30 to 40 cm, with a maximum of 80 cm depending on the month and the time between two visits. Every 10 to 15 days, the instrument was re-adjusted manually to a set height of 2 m. More details on the sensors, the field campaign and data processing are available in Litt *et al.* (2017).

### 2.3.3 Raw SAFRAN reanalysis data

Since AWS records on glaciers are limited in time and scarcely distributed, the near surface meteorological forcing data are estimated by meteorological reanalyses. In this study, we used the SAFRAN meteorological re-analysis system (Durand *et al.,* 2009). SAFRAN data are provided using atmospheric vertical profiles simulated by an atmospheric model (ERA-40 reanalysis until 2001 and ARPEGE operational model after 2002). Results are then corrected by optimal interpolation with observed meteorological data from various sources (automatic weather stations, manual observations carried out in the climatological network or at ski resorts, remotely-sensed cloudiness, atmospheric upper-level sounding). Note that surface observations that could be used to correct data are scarce at very high altitudes (*i.e.* above 2000 m a.s.l.).

SAFRAN outputs include hourly meteorological variables (2-m air temperature and relative humidity, precipitation amounts and phases, incoming direct and diffuse shortwave radiation, incoming longwave radiation, wind speed, cloudiness) that are assumed to be homogeneous within a given massif (in particular within the Grandes Rousses massif where the Saint-Sorlin Glacier is located, Figure 1a) and depend only on altitude (one data every 300 m) and aspect (7 orientations available: N, NE, NW, S, SW, SE and 'Flat'). The direct solar radiation is provided for an infinite flat area but can be easily projected for

any aspect and slope (Lafaysse *et al.,* 2011) using the Crocus model (see section 3.1.1). However, as the surface slope angles were small at each station and because the radiometers were not found to be far from horizontal during field visits, no slope correction was applied to the measured shortwave radiation. Shading from surrounding topography is taken into account in the computation of shortwave radiation, but the impact of emitted long wave radiation and reflected short wave radiation by surrounding slopes is not considered.

SAFRAN outputs are available in 300-m elevation steps. In our study, they were linearly interpolated (following the vertical and horizontal axes) on the 200-m horizontal resolution grid encompassing the glacier.

| Station | Location | Date of records | Timestep | Variables | Instrument | Manufacturer accuracy | Associated studies |
|---|---|---|---|---|---|---|---|
| AWS$_m$ | Moraine 2720 m a.s.l. | 2005-present | 30 min | Aspirated air T (°C) Relative humidity (%) Wind speed (m s$^{-1}$) and direction (degrees) Upward SW (W m$^{-2}$) Downward LW (W m$^{-2}$) | Vaisala HMP45C Vaisala HMP45C Young 05103 Young 05103 Kipp and Zonen CG3 Kipp and Zonen CG3 | ±0.2°C 3% 0.3 m s$^{-1}$ ±3° 0.4% 0.4% | *Six et al.* (2009) *Sicart et al.* (2008) *Dumont et al.* (2012) |
| AWS$_{g06}$ | Ablation area 2770 m a.s.l. | 9 July - 28 August 2006 | 30 min | Aspirated air T (°C) Relative humidity (%) Wind speed (m s$^{-1}$) and direction (degrees) Upward SW (W m$^{-2}$) Downward LW (W m$^{-2}$) EC measurements | Vaisala HMP45C Vaisala HMP45C Young 05103 Young 05103 Kipp and Zonen CG3 Kipp and Zonen CG3 Csat3 and Licor 7500 | ±0.2°C 3% 0.3 m s$^{-1}$ ±3° 0.4% 0.4% 0.4% | *Dumont et al.* (2012) *Litt et al.* (2016) |
| AWS$_{g-accu08}$ | Accumulation area 2900 m a.s.l. | 12 July - 10 September 2008 | 30 min | Aspirated air T (°C) Relative humidity (%) Wind speed (m s$^{-1}$) and direction (degrees) Upward SW (W m$^{-2}$) Downward LW(W m$^{-2}$) | Vaisala HMP45C Vaisala HMP45C Young 05103 Young 05103 Kipp and Zonen CG3 Kipp and Zonen CG3 | ±0.2°C 3% 0.3 m s$^{-1}$ ±3° 0.4% 0.4% | *Dumont et al.* (2012) |
| AWS$_{g08}$ | Ablation area 2770 m a.s.l. | 11 July - 2 August 2008 | 30 min | Aspirated air T (°C) Relative humidity (%) Wind speed (m s$^{-1}$) and direction (degrees) Upward SW (W m$^{-2}$) Downward LW (W m$^{-2}$) | Vaisala HMP45C Vaisala HMP45C Young 05103 Young 05103 Kipp and Zonen CG3 Kipp and Zonen CG3 | ±0.2°C 3% 0.3 m s$^{-1}$ ±3° 0.4% 0.4% | *Dumont et al.* (2012) |
| AWS$_{g09}$ | Ablation area 2770 m a.s.l. | 13 June - 4 September 2009 | 30 min | Aspirated air T (°C) Relative humidity (%) Wind speed (m s$^{-1}$) and direction (degrees) Upward SW (W m$^{-2}$) Downward LW (W m$^{-2}$) Albedo | Vaisala HMP45C Vaisala HMP45C Young 05103 Young 05103 Kipp and Zonen CG3 Kipp and Zonen CG3 | ±0.2°C 3% 0.3 m s$^{-1}$ ±3° 0.4% 0.4% | *Dumont et al.* (2012) *Litt et al.* (2016) |

**Table 1.** AWS sensor characteristics, locations and measurement periods. Slope angles and aspects are similar for all stations located on the glacier (AWSg06, AWSg-accu08, AWSg08, AWSg09), respectively 5° and roughly North. Slope at AWSm is 0°.

### 2.3.4 Adjusted SAFRAN data

SAFRAN data were compared to the AWS$_m$ measurements over 10 years (2005-2015) and to the available AWS$_g$ measurements. Biases were adjusted and the influences of all corrections mentioned below on the simulated SMB are discussed in section 4.2. SAFRAN and AWS$_m$ hourly air temperatures over the ablation and accumulation seasons are well correlated ($R^2$ = 0.98 (summer) and 0.99 (winter), both significant at the 99% confidence level (Student's *t* test), and the

Root-Mean-Square Errors (RMSE) are 0.7°C (summer) and 0.76°C (winter)). Hourly SAFRAN relative humidity is also in good agreement with the $AWS_m$ data ($R^2$ = 0.74, significant at the 95% confidence level, and RMSE = 13.6%). The comparison between SAFRAN and $AWS_m$ incoming long wave radiation indicates an overestimation of SAFRAN data for low cloudiness conditions. This can be caused by high-altitude clouds that are not considered in SAFRAN reanalysis and an incorrect vertical discretization of the atmosphere in SAFRAN. As proposed by Dumont *et al.*

(2012), we corrected the long wave incident radiation (LW in W $m^{-2}$) by implementing a linear function depending on SAFRAN cloudiness (ranging from 0 to 1) (Eq. 1):

$$LW_{corrected} = LW_{SAFRAN} - (a*Cloudiness + b) \quad (1)$$

where a = -0.56 and b = 38 W $m^{-2}$ are empirical parameters, calibrated with $AWS_m$ measurements. This correction was

calibrated over the 2005-2015 period and applied over the 1995-2015 period. Using this correction, the correlation between $AWS_m$ incoming LW radiation and corrected LW radiation from SAFRAN increased the correlation from $R^2$ = 0.71 to $R^2$ = 0.83 and the RMSE decreased from 44.3 W $m^{-2}$ to 29.7 W $m^{-2}$. Correlations between daily incoming shortwave radiation ($R^2$ = 0.81) are significant at the 99% confidence level (Student's t test) and RMSE = 77.2 W $m^{-2}$.

A poor correlation ($R^2$ = 0.19, RMSE = 3.8 m $s^{-1}$) between SAFRAN wind speed (considered at 2-m) and measured values at $AWS_m$ (at ~2-m) is observed and is mainly due to an underestimation of strong winds by SAFRAN. Differences between $AWS_m$ and SAFRAN wind speed range from 0.9 to 21 m $s^{-1}$ with a mean value of 4.3 m $s^{-1}$. This underestimation is likely due to both non-consideration of katabatic wind and local effects due to orography (Dumont *et al.*, 2012). As mentioned in Litt *et al.* (2017), when large-scale atmospheric forcing was strong, intense downslope winds were observed, aligned with

the main glacier flow (*i.e.* coming from the South, see Figure 1). The wind speed measured at $AWS_m$ (glacier foreland) were first compared to the wind measured at $AWS_{g06}$ and $AWS_{g09}$. Since the correlation between the measured wind speed on the foreland and on the glacier is high ($R^2$=0.97, RMSE=1.7 m $s^{-1}$), we assumed the wind speed measured at $AWS_m$ to be representative at the glacier scale and used it to replace SAFRAN wind speed estimates in this study. However, data is limited to the 2005-2015 period. Outside this period (over 1995-2004), the SAFRAN wind speed was corrected using a

quantile-mapping method (Déqué, 2007; Gobiet *et al.*, 2015). This method was chosen because it is considered to be one of the most efficient bias adjustment methods available (e.g., Gobiet *et al.*, 2015). Percentiles of the observed distribution ($AWS_m$ measurements) and the SAFRAN distribution are calculated using every data of a given month and for each month over the 2005-2015 period. A linear method was used for mapping and extrapolated data over the minimum/maximum observed quantile were estimated with a linear function. The resulting mapping function of the quantile-quantile plot was

used to adjust the SAFRAN wind speed distribution over the 1995-2004 period.

Finally, SAFRAN cumulated winter precipitation over each winter was compared to the winter SMBs measured at each

accumulation measurement site. As already mentioned in previous studies (Gerbaux *et al.,* 2005; Dumont *et al.,* 2012), using SAFRAN raw data leads to a significant underestimation of the winter SMB. The accumulation amount was adjusted based on the methodology developed in previous studies (Vincent, 2002; Gerbaux *et al.,* 2005; Dumont *et al.,* 2012; Réveillet *et al.,* 2017). For each winter, individual winter SMB measurements were first used to compute multiplication factors for SAFRAN precipitation. The multiplication factors were then spatially interpolated over the entire glacier surface area (kriging method) to obtain an annual map of multiplicative factors. These factors were then used to correct solid and liquid precipitation. The factors varied from 1.2 to 2.1 depending on both the year and the site. Applying these factors led to an increase in winter SMB ranging from 0.05 m w.e. yr$^{-1}$ to 1.64 m w.e. yr$^{-1}$ depending on the site (with a mean of 0.46 m w.e. yr$^{-1}$).

All adjustments of the raw SAFRAN data described below are summarized in Figure 2. The impact of these corrections on the simulated SMB is discussed in section 4.2.2.

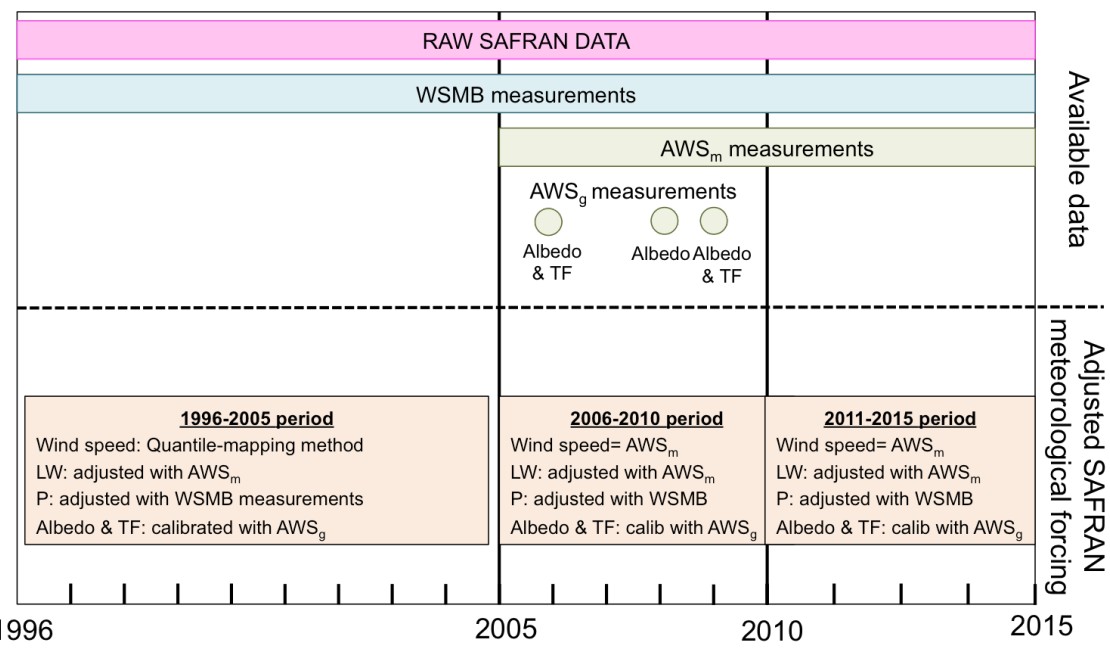

**Figure 2.** Summary of available meteorological data and the adjustments of the raw SAFRAN data, depending on the study period (TF = Turbulent Fluxes).

## 3 Methodology: model descriptions and evaluation metrics

### 3.1 Model descriptions

#### 3.1.1 Crocus model

The Crocus snowpack model, implemented as one of the snow scheme options of the SURFEX/ISBA land surface model (Masson *et al.,* 2013), was originally developed by Météo-France to simulate seasonal snowpack and to assist in avalanche hazard forecasting over the French mountain ranges (Brun *et al.,* 1989; Vionnet *et al.,* 2012). Crocus is a full energy balance, one-dimensional snowpack model, driven by meteorological variables including temperature, shortwave radiation, longwave radiation, specific humidity, rainfall and snowfall rates and wind speed. It simulates a layered snowpack with a Lagrangian representation, each layer being characterized by its thickness, density, temperature, liquid water content and two semi-empirical variables to describe the snow/ice microstructure. The variables are grain size/dendricity and sphericity (see Vionnet et al., 2012, for more details). Their values are specified for glacier ice. The specified values only impact the calculation of albedo/light penetration depth, which is constant for ice. The number of numerical snow layers evolves with time to tend towards an idealized prescribed thickness profile that is appropriate for the computation of an accurate energy balance (thinner layers close to the surface) but that avoids the aggregation of snow layers with different microstructural properties. The model solves the heat diffusion equation in the snowpack at a 15-minutes time step considering the different energy fluxes between the surface and the atmosphere and between the bottom of the snowpack and the soil. Physical processes such as solar radiation absorption, liquid water percolation, snow metamorphism and settlement are also considered by the model. The snowpack model can be used on icy surfaces, considering an ice layer as a specific snow layer with a density of 917 kg m$^{-3}$ (Gerbaux *et al.,* 2005; Lejeune *et al.,* 2007; Dumont *et al.,* 2012). The specific parameterizations used in our study (albedo and roughness length) will be described in detail below. A more general presentation of Crocus can be found in Brun *et al.* (1992) and Vionnet *et al.* (2012).

In the initial version of Crocus, solar radiation is handled in three separate spectral bands ([0.3-0.8], [0.8-1.5] and [1.5-2.8] μm), and albedo is computed for each band as a function of the snow properties: grain size, shape and age (Brun *et al.*, 1992). In this initial version, snow albedo ranges from 1 to 0.7 in the UV and visible range ([0.3-0.8] μm) and depends on the optical diameter and on the amount of light absorbing impurities, the latter being parameterized with respect to the age of snow (with a time constant of 60 days). In our study, the minimum snow albedo is set to 0.5 to consider older snow with higher impurity content (Cuffey and Patterson, 2010) and the time constant for the impurities parameterization is reduced to 20 days. In particular, firn albedo is considered as old snow albedo. Ice albedo is constant with time for all the considered spectral bands. Values are set to [0.23, 0.16, 0.05], based on previous studies on Saint-Sorlin Glacier (Gerbaux *et al.*, 2005; Dumont *et al.*, 2012). Note that albedo measurements performed at AWS$_{g06}$, AWG$_{g08}$ and AWS$_{g-accu08}$ were used to calibrate and validate ice and snow albedo in the model (see section 4.2.3.2).

In Crocus, the sensible and latent heat fluxes (respectively H and LE) are calculated using the bulk aerodynamic approach, including a stability correction (Brutsaert, 1982). The two fluxes are parameterized using an effective surface roughness length $z_0$ (Vionnet *et al.*, 2012), with different values for snow and ice surfaces. Note that this roughness length $z_0$ is considered as an effective value used in the model to fix the aerodynamic ($z_m$), temperature ($z_t$) and humidity ($z_q$) roughness values, following the approximation: $z_0 = z_m = 10z_t = 10z_q$. The choice of appropriate values for $z_0$ over ice ($z_{0ice}$) for Saint-Sorlin Glacier is presented in section 4.2.3.1. As no turbulent flux measurements are available for the snow surface, the snow

roughness length ($z_{0snow}$) is arbitrarily fixed at 0.1 mm (Gromke *et al.*, 2011).

### 3.1.2 Temperature-index model

The empirical model selected in this study is the ATI (Alternative Temperature-Index) model proposed by Réveillet *et al.* (2017). In this approach, the daily melt is computed as follows:

$$M = Tf_{ice/snow} * T + If_{ice/snow} * IPOT \qquad (2)$$

where $Tf_{snow/ice}$ is the temperature factor (m w.e. $d^{-1} °C^{-1}$) which depends on the surface condition (*i.e.* ice or snow), $T$ is the mean daily air temperature (°C), $If_{snow/ice}$ is the radiation factor ($m^3$ w.e. $d^{-1} W^{-1}$) which also depends on the surface condition

(*i.e.* ice or snow) and *IPOT* is the potential clear-sky direct solar radiation (W $m^{-2}$) calculated following Hock (1999). Melt can occur when the sum of the two terms of the equation is positive, meaning that melt can occur even if T is <0°C. In this approach, $If_{snow/ice}$ represents the energy fluxes related to solar radiation, which differ for snow and ice, but are assumed constant in time (*i.e.* no temporal change in the albedo of the snow or ice is taken into account). *Tf* represents the temperature-dependent energy fluxes such as turbulent fluxes or LW radiation. Empirical factors were calibrated with

punctual SMB measurements performed on Saint-Sorlin Glacier over the period 1995-2012 (more details on the model and the calibration can be found in Réveillet *et al.*, 2017).

### 3.2 Evaluation metrics

### 3.2.1 Model evaluation method

The Crocus model was applied over the 1995-2015 period and evaluated over three distinct time periods, depending on the available AWS measurements (Figure 2): (i) a calibration period (2006-2010), over which it was possible to correct both meteorological forcing and model parameterization (albedo and roughness length) using $AWS_g$ and $AWS_m$ measurements, (ii) the 2011-2015 period over which it was possible to correct only meteorological forcing using $AWS_m$ measurements, and finally (iii) the 1996-2005 period over which no corrections were possible, due to the absence of AWS measurements.

Results of annual, winter and summer mass balance simulation using Crocus are presented section 4.1.1.

Crocus model simulations were then compared to those obtained from the ATI temperature-index model. The ATI model was forced with the same winter SMB simulated by Crocus, to compare the ability of the two models to simulate summer SMB only. Note that in the ATI model, summertime snowfalls are deduced from SAFRAN data. Comparisons were performed over two periods: (i) the period for which AWS measurements were available (2006-2015) and (ii) the period

without AWS measurements available (1996-2005).

Simulations were performed with a 200-m DEM resolution (see sections 2.2.2 and 2.3.3) and grid cells corresponding to stake locations were extracted for comparison between modelled and measured SMBs. Note that a 200 m resolution was chosen as a compromise to be sufficiently precise to consider the spatial variation of Saint Sorlin glacier (in particular the variation of aspect) and capture variability between stakes, while maintaining relevance regarding the meteorological forcing

(given that values are available every 300 m of elevation). Performance was evaluated by comparing summer SMB simulated by the ATI and Crocus models to summer SMB measurements of each stake located in the ablation and accumulation areas. Note that comparisons were made over the exact same period, determined by SMB measurement dates. The results are presented in section 4.2.3.

Finally, the sensitivity of annual SMB to both winter and summer SMB was assessed using the Crocus model at various

stakes in the ablation area. First, we considered averaged winter conditions over the accumulation period (from 1 October to 15 April) by computing the average of the 20 available winters (1996 to 2015). Then, based on this averaged winter, 20 simulations of annual SMB were performed using each of the 20 summer conditions (1996-2015).

Next, we assessed the sensitivity of annual SMB to winter SMB. We considered an averaged summer by computing the mean of the SAFRAN corrected re-analysis of the twenty summers available (1996-2015). Simulations were performed

using the twenty winter conditions available. The results are presented in section 4.1.3.

### 3.2.2 Analysis of SMB sensitivity to Crocus parameterization

#### 3.2.2.1 DEM

First, we investigated the effect of the spatial resolution of the DEM. For this purpose, the numerical simulations were

performed with a 50-m resolution grid size, based on the 2007 DEM, and were compared with the results obtained using the same DEM with a 200-m resolution grid. Second, the impact of changes in glacier surface topography with time was evaluated by performing simulations over the 2006-2010 period using the three DEMs (1998, 2007 and 2014). To evaluate these sensitivities, summer SMBs simulated by Crocus were compared to summer SMB measurements at each stakes and the results are presented in section 4.2.1.


#### 3.2.2.2 Meteorological forcing

To test the impact of the correction made on the longwave radiation, wind speed and precipitation, simulations were performed using a raw SAFRAN forcing and the adjusted SAFRAN forcing described in section 2.3.4. Evaluation involved comparing SMBs simulated by Crocus with SBMs measured at each stakes, over the 2006-2010 period. The results are

presented section 4.2.2. Regarding the precipitation, two additional adjustment methods were used. The first is based on the use of a single mean correction factor, computed using all available winter SMBs (over the 1996-2015 period). The second method is based on the use of a temporally averaged spatialized map of multiplicative factors based on the twenty years of available measurements (as proposed by Gerbaux *et al.,* 2005 and Dumont *et al.,* 2012).

**3.2.2.3 Crocus parameters**

In the Crocus version used in this study, both surface roughness and albedo were calibrated using AWS measurements. Sensitivity tests were performed by varying these variables to estimate the uncertainties when no measurements are available.

The effective roughness length values were varied arbitrarily by a factor of 1 to 100 and the ice albedo of the spectral band [0.3-0.8] μm were varied from 0.16 to 0.32 (in agreement with Oerlemans *et al.,* 2009). Simulations were performed at different stakes for the 2006-2010 period. The results are presented section 4.3.3.

## 4. Results and discussion

### 4.1 Surface mass balance modelling

#### 4.1.1 Crocus performance

The Crocus model was run over the three distinct time periods and annual and seasonal SMBs were compared to measurements (Figure 3). Correlations are significant in every case at the 95% confidence interval according to a Student's *t* test.

Performance over both the period 2006-2010 and the recent period 2011-2015 is similar. Winter SMB correlations for the recent period are high (Nash and Sutcliffe coefficient (NS) > 0.72, (Figures 3e and 3h), Nash and Sutcliffe (1970)). This high performance results from the use of annual multiplication factors to correct precipitation to fit with accumulation measurements. As a consequence, differences between measured and simulated winter SMBs (systematically lower than 0.5 m w.e.) are due to the interpolation method and some melting events which can occur over the accumulation period. For these two periods (2006-2010 and 2011-2015), summer SMB simulations were also in good agreement with measurements (NS > 0.85) in both accumulation and ablation areas (Figures 3f and 3i), indicating good performance of the model in simulating SMB changes over the ablation season. Due to both good winter SMB and summer SMB simulations, results at an annual scale (Figures 3d and 3g) also showed the good performance of the model (NS > 0.67).

Regarding the period 1996-2005 (Figure 3, a-b-c), while correlations between measured and simulated SMBs are significant at the 95% confidence interval according to a Student's *t* test, results indicate lower performance, especially for the simulation of the summer SMBs (Figure 3c). Simulated summer SMBs and annual SMBs (Figures 3a and 3c) are over-estimated for very negatives summer SMBs observed in 2002/2003 in the ablation area.

#### 4.1.2 Comparison with the temperature-index approach

Over the period 2006-2015, results indicate better performance with the Crocus model (Figures 4a and b). Indeed, the ATI model under-estimated the summer SMB values, when observed summer SMB is above -2 m w.e., and in particular those corresponding to the accumulation area (Figure 4a)). This leads to a significant decrease in the correlations between measurements and simulations. However, when considering summer SMB measurements in the ablation area only, performance is similar for the two models (NS is 0.47 for Crocus and 0.51 for the ATI model).

In addition, the temporal evolution of the simulated summer SMBs over season is shown in Figure 5. Daily summer SMB data simulated by both models are reported in each graph for different measurement points. Note that this was done for all

years over the period 2006-2015, but only results for the year 2008 are represented here for the sake of clarity. For this year, results indicate very similar performance for the two models in the lower part of the ablation area (stakes 1 to 22) at the end of the season: the absolute mean difference of summer SMB is 0.13 m w.e. yr$^{-1}$ (lower than the measurement uncertainty) and the maximum difference is 0.36 m w.e. yr$^{-1}$. The same is true when we consider all the ablation season results (i.e. 2006-2015): the maximum difference of 0.45 m w.e. yr$^{-1}$. For the year 2008, the ATI model simulates lower ablation compared to Crocus model during August in the lowest part of the glacier (e.g. stakes 20 and 22), and higher ablation during October (Figure 5). However these results are specific to this year and these stakes. No systematic difference is observed.

In the accumulation area and close to the equilibrium line (e.g. stakes 27, 30), differences between the summer SMBs simulated by the two models (Figure 5) are greater: the absolute mean difference of summer SMB is 0.56 m w.e. yr$^{-1}$ and the maximum difference is 0.87 m w.e. yr$^{-1}$. Considering all the years, the maximum difference is 1.91 m w.e. yr$^{-1}$. Here again, there is no systematic difference, except that maximum differences are generally observed in June and October.

Over the period 1996-2005, considering all the point data over the entire glacier, Crocus performs better than the ATI model (Figures 4c and d). Here again, summer SMBs simulated with the ATI model in the accumulation area are under-estimated. On the other hand, when considering the ablation area only, results from the ATI model better fit the summer SMB measurements (NS is 0.36 for Crocus and 0.59 for the ATI model). Decreasing Crocus performance over the 1996-2005 period can be explained by the absence of AWS measurements to evaluate and validate the correction made on the wind speed and longwave forcing data.

Note that the ATI was calibrated over the period 2005-2015 and is stable over the 20 years of simulations, considering an uncertainty of 0.2 m w.e. (Réveillet *et al.,* 2017). However, stability of the parameters over a period of more than two decades cannot be guaranteed.

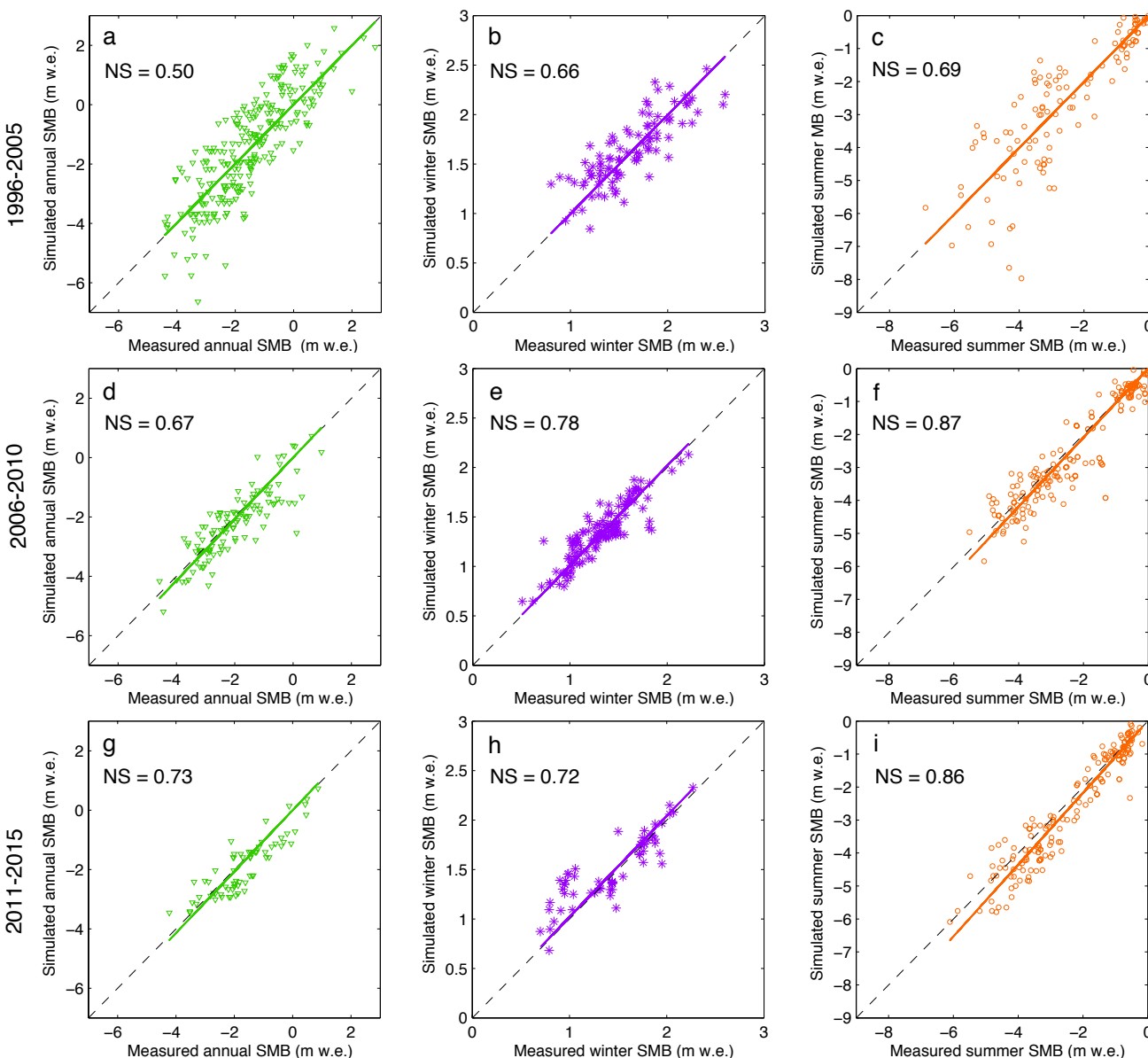

**Figure 3.** Comparisons between simulated and measured SMBs (m w.e.) at each measurement point over the 1996-2005 (*in-situ* meteorological measurements not available; a, b, c), 2006-2010 (*in-situ* meteorological measurements available on the moraine and on the glacier; d, e, f) and 2011-2015 (*in-situ* meteorological measurements available on the moraine only; g, h, i) periods. The annual (a, d, g), winter (b, e, h) and summer (c, f, i) SMBs are shown in green, purple and orange, respectively. The Nash and Sutcliffe coefficient (NS) is indicated on each graph.

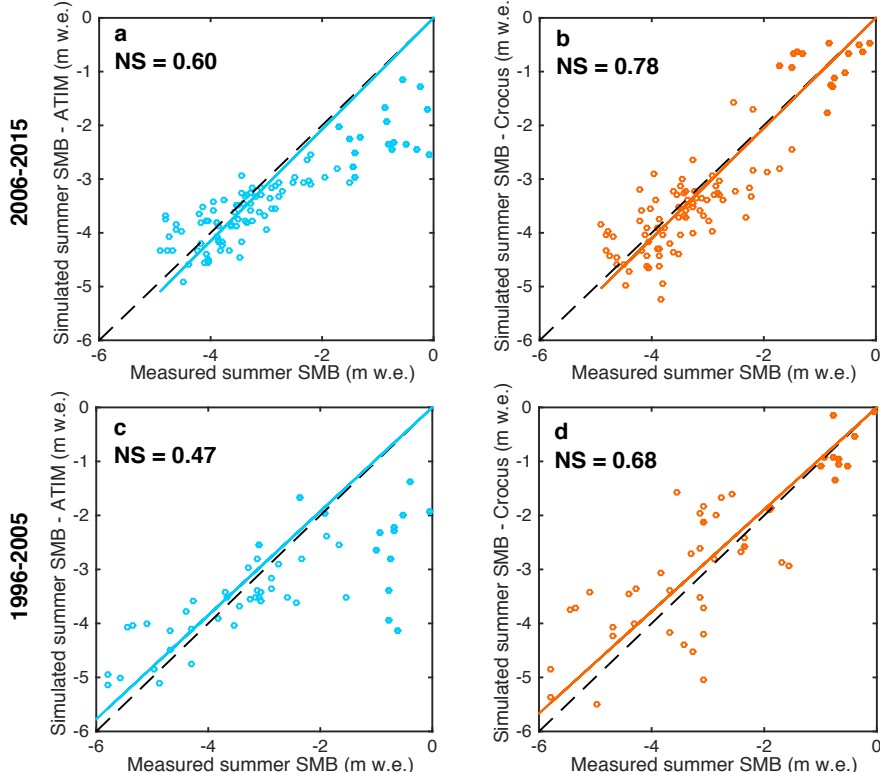

**Figure 4.** Correlations between simulated (blue (a and c) for the ATI model and orange (b and d) for the Crocus model) and measured summer SMBs at each stake of Saint-Sorlin Glacier over the 2006-2015 period (a and b) and the 1996-2005 period (c and d). Circles represent measurements in the ablation area and solid dots represent measurements in the accumulation area.

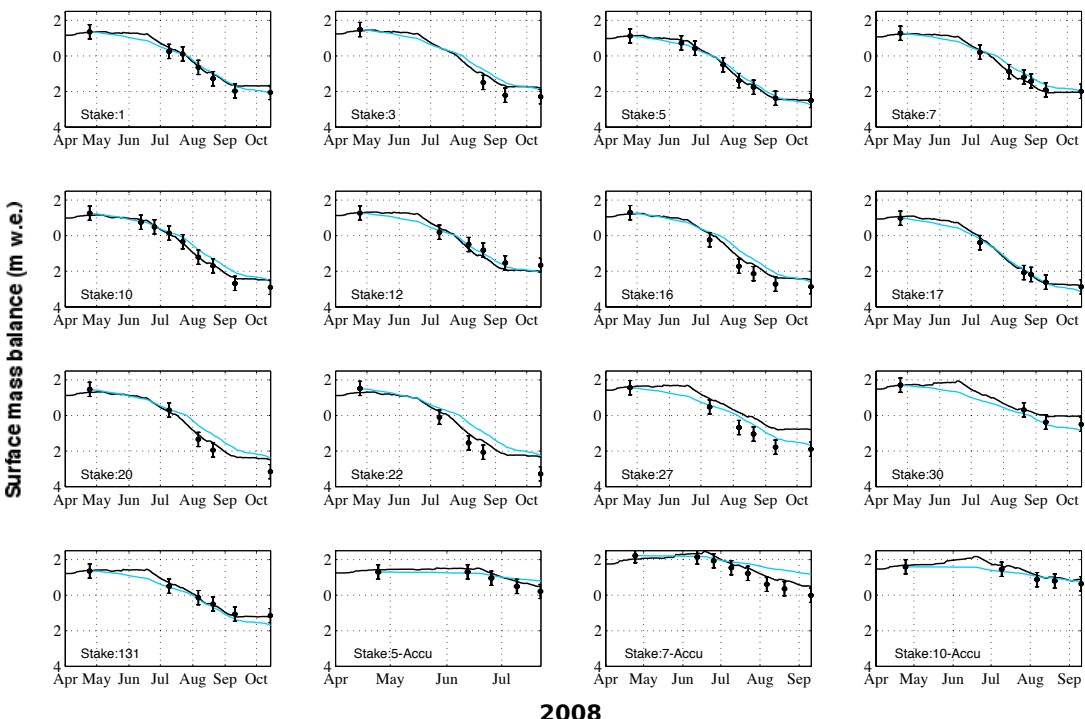

**2008**

**Figure 5**. Surface mass balance evolution at some selected measurement points of Saint-Sorlin glacier over the hydrological year 2007-2008. Black lines represent the simulated SMB using Crocus model with corrected forcing. The blue curves show the simulation made with the ATI model using simulated winter SMB adjusted with measurements. Black dots represent the measurements with their uncertainties.

**4.1.3 Annual mass balance sensitivity to seasonal mass balance**

The tests (described in section 3.2.1) of the annual mass balance sensitivity to seasonal mass balance using the Crocus model were performed at seven stakes in the ablation area, ranging between 2700 m a.s.l. and 2870 m a.s.l. For the sake of clarity, only the results for stake #10 (located at 2760 m a.s.l.) are presented in Figure 6, but conclusions are similar for all the stakes.

Regarding the sensitivity of annual SMB to summer SMB (Figure 6a), the results show that the simulated annual SMB was the least negative with 1995 summer conditions (green curve) and the most negative with 2003 summer conditions (red line). The difference in annual SMBs between these two extreme summers for the stake #10 was 4.1 m w.e. yr$^{-1}$ at the end of the hydrological year. Similar results are found for the other stakes: the mean difference is 4.4 m w.e. yr$^{-1}$ with a standard deviation of 0.41 m w.e. yr$^{-1}$.

The sensitivity of annual SMB to winter SMB is illustrated by Figure 6b. Note that for the sake of clarity, only the two extreme years of the time series (2000-2001, highest winter SMB (pink line) and 2008-2009, lowest winter SMB (blue line)) are presented in Figure 6b. The difference in annual SMBs between these two extreme summers for the stake #10 was 4.1 m

w.e. yr$^{-1}$ at the end of the hydrological year. Similar results are found for the other stakes: the mean difference is 4.4 m w.e. yr$^{-1}$ with a standard deviation of 0.41 m w.e. yr$^{-1}$.

The sensitivity of annual SMB to winter SMB is illustrated by Figure 6b. Note that for the sake of clarity, only the two extreme years of the time series (2000-2001, highest winter SMB (pink line) and 2008-2009, lowest winter SMB (blue line)) are presented in Figure 6b. The difference between these two years on 15 April is 1.2 m w.e at stake #10 (and on average 1.1 m w.e with a standard deviation of 0.13 m w.e considering all the stakes). Using the same summer conditions, the difference at the end of the hydrological year is 2.4 m w.e. (*i.e.* twice the difference at the end of the winter season). Here again results are similar for the all the stakes considered: the mean difference is 2.2 m w.e. yr$^{-1}$ with a standard deviation of 0.21 m w.e. yr$^{-1}$.


The same test was performed using the extreme 2003 summer conditions instead of the mean summer conditions. In this case, the difference at the end of the hydrological year for all the stakes was considerably larger (3.4 m w.e., in mean, standard deviation 0.45 m w.e. yr$^{-1}$; results not shown). These results confirm that the annual SMB variability is mainly driven by the summer SMB variability (*i.e.* differences are larger when we considered a mean winter and all the summer conditions than

the contrary). Nevertheless, the annual SMB appears to be very sensitive to the winter SMB, in particular for extreme years.

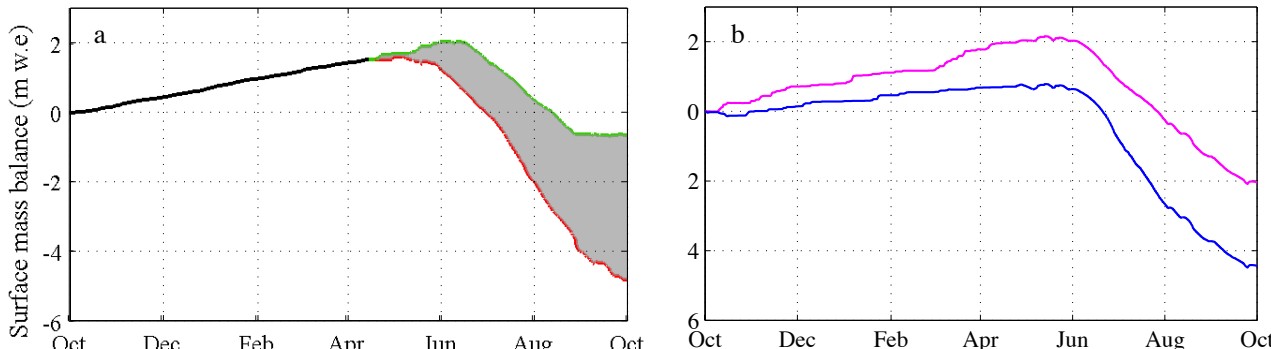

**Figure 6.** (a) Surface mass balance at stake #10, (2760 m a.s.l.) over one hydrological year, using averaged winter conditions and all summer conditions for the period 1995-2015. Red line represents simulation with 2003 summer conditions and the green line represents 1995 summer conditions. All the other years are included in the grey area. (b) Surface mass balance at

stake #10, over one hydrological year, using averaged summer conditions (over 1995-2015), 2000-2001 winter conditions (pink) and 2008-2009 winter conditions (blue), representing the two extreme results.

### 4.2 Sensitivity of SMB to Crocus parameterization

### 4.2.1 Digital elevation model resolution and date

Regarding the effect of the spatial resolution of the DEM (*i.e.* 50-m *vs* 200-m resolution grid), changes in winter SMB are

negligible (NS coefficients are equal). Surprisingly, our results also indicate similar performance in simulating the summer SMB when using a 50-m or 200-m resolution DEM (not shown here), even if changing the resolution impacts the calculation of slope and aspect and affects the incoming radiation computation (shadowing effect).

On the other hand, the comparison between the 1998 and 2014 DEMs shows surface elevation lowering ranging from 0 to -52 m and an average slope increase from 0 to 6°, with larger slope changes found in the ablation area. The impact of these changes was evaluated for different areas. First, correlations between simulated and measured summer SMB were computed for all the stake measurements (in the accumulation and ablation areas), then for the stakes located in the ablation area only and finally for the stakes located in the lower part of the glacier tongue. The differences between the simulated and measured

summer SMBs are reported in Table 2 (correlations are not statistically different). The highest differences between simulations and measurements are obtained for the stakes located in the lower part of the glacier tongue, using 1998 and 2007 DEMs (*i.e.* where geometric changes are the greatest). Simulations performed with 1998 and 2007 DEMs led to a mean difference in simulated summer SMBs of 0.19 m w.e. yr$^{-1}$ (~5% of the SSMBs) and reached 0.64 m w.e. yr$^{-1}$ for the lowest stakes (~15% of the summer SMBs and ~20% of the annual SMBs). Simulations performed with 2007 and 2014 DEMs, led

to a mean difference of 0.15 w.e. yr$^{-1}$ (<5% of the SSMBs) and a maximum of 0.47 5 w.e. yr$^{-1}$ for the lowest stakes. Note that the differences in simulated summer SMBs *vs.* measurements in the accumulation area are larger when considering the DEMs from 2014 and 2007 than with 1998 and 2007 DEMs and can reach 0.38 m w.e. yr-1 (~20% of the summer SMBs and ~25% of the annual SMBs). Despite changes in glacier surface topography over the entire study period, such changes only affect the simulated summer SMB (*i.e.* considering changes larger that measurement uncertainty) for a limited number

of individual stakes (maximum 5). Considering the entire glacier, these changes in the simulated summer SMB are negligible as the mean is lower than the measurement uncertainty.

| DEM date | NS (all stakes) | NS (stakes of the ablation area) | NS (stakes close to the tongue) |
|---|---|---|---|
| 1998 | 0.87 | 0.41 | 0.79 |
| 2007 | 0.87 | 0.42 | 0.85 |
| 2014 | 0.86 | 0.47 | 0.82 |

**Table 2.** NS efficiency coefficient for simulated summer mass balances with respect to measured values over the 2006-2010 period using different 200-m resolution DEMs. The evaluation was performed using all stake measurements, only stakes

located in the ablation area and stakes located in the tongue of the glacier where geometry changes are larger.

**4.2.2 Meteorological inputs**

An important question is whether the Crocus model forced with SAFRAN reanalysis data could be used on a large set of

glaciers or over a long time period without *in-situ* meteorological measurements available to evaluate or correct the atmospheric forcing. The sensitivity of the model to the corrections made on the meteorological forcing described in section 2.3.4 and summarized in Figure 2 is presented below. Uncertainties are calculated over the 2006-2010 period, at each

measurement point of the glacier.

### 4.2.2.1 Sensitivity to precipitation correction

SAFRAN precipitation was corrected annually using an extensive data set of winter SMBs on Saint-Sorlin Glacier. Here we test different approaches to correct SAFRAN precipitation to consider the case when such extensive measurements are not available.

First, as already mentioned in previous papers (e.g., Gerbaux *et al.,* 2005; Dumont *et al.,* 2012),  using raw SAFRAN precipitation leads to an underestimation of the winter SMB due to the lack of observations in high altitude areas and the complexity of considering local effects such as wind transport. Using raw SAFRAN precipitation data leads to a very low NS coefficient for simulated winter SMBs with respect to observed values. This difference in terms of winter SMB also strongly impacts the performance in simulating summer SMB and annual SMB (Table 3).

Second, based on the results provided by a method using a single mean correction factor over the entire glacier surface area, equal to 1.73 for Saint-Sorlin Glacier, there is a significant decrease in the correlation between measured and simulated winter SMBs and lower performance in the simulation of summer SMBs.

Finally, the use of an averaged spatialized map of multiplicative factors also showed a decrease in the efficiency of both winter and summer SMB estimates (NS decreased from 0.78 to 0.15 and from 0.87 to 0.77 respectively).

These results suggest that, for Saint-Sorlin Glacier, the accuracy of the seasonal SMB computation is affected by the spatial and temporal aspects of the precipitation adjustment. This highlights the importance of considering local effects driving the spatio-temporal variability of the winter SMB, such as wind transport and sublimation.

| | *NS* for ASMB | *NS* for WSMB | *NS* for SSMB |
|---|---|---|---|
| Annual map of factors | 0.67 | 0.78 | 0.87 |
| No adjustment | -0.01 | -0.03 | 0.23 |
| Constant factor : 1.73 | 0.47 | 0.09 | 0.75 |
| Mean factors on 1996-2015 period | 0.49 | 0.15 | 0.77 |

**Table 3.** NS efficiency coefficients for simulated surface mass balances with respect to measured values over the 2006-2010 period. Simulations were performed using three different approaches to correct precipitation and were evaluated for the winter SMB, summer SMB and annual SMB.

### 4.2.2.2 Sensitivity to incoming longwave radiation

The impact of the incoming longwave radiation corrections is significant and considerably affects the simulated summer SMB. A good example is the 2007-2008 hydrological year shown in Figure 7 (pink curves). Because the SAFRAN raw incoming longwave radiation is over-estimated for low cloudiness conditions (by about 30%), the correction leads to a decrease in the energy available for melt and thus a less negative summer SMB. Simulations performed with and without longwave correction indicate a mean difference (computed with all available measurements over the 2006-2010 period) at the end of the season of 0.54 m w.e. yr$^{-1}$ (with a standard deviation equal to 0.60 m w.e. yr$^{-1}$). Hence, not considering the

incoming longwave radiation correction leads to a significant decrease in the NS coefficient (see Table 4). Note also that Sautner and Obleitner (2015) found a high sensitivity of the Crocus snowpack model to errors of incident longwave radiation over glaciers in Svalbard.

| | $NS$ for ASMB | $NS$ for WSMB | $NS$ for SSMB |
|---|---|---|---|
| SAFRAN with corrected data | 0.67 | 0.78 | 0.86 |
| Without LW correction | 0.36 | 0.73 | 0.65 |
| Without wind speed correction | 0.27 | 0.59 | 0.71 |

**Table 4.** NS efficiency coefficients for simulated surface mass balances with respect to measured values over the 2006-2010 period. Simulations were performed with and without correction of the meteorological forcing from SAFRAN.

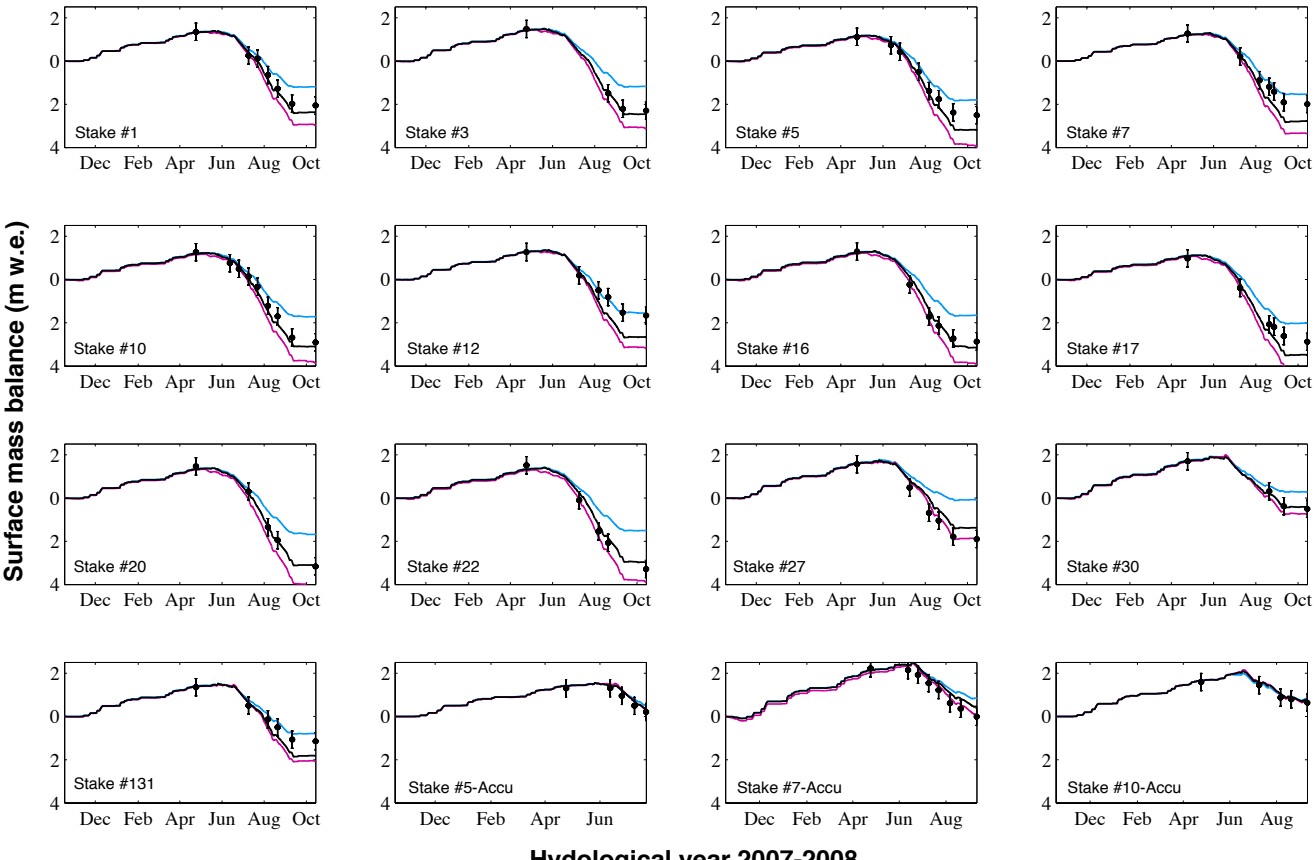

**Figure 7.** Surface mass balance (in m w.e.) at some selected measurement points in the accumulation (Accu) and ablation areas of Saint-Sorlin Glacier (numbers refer to the stake numbers shown in Figure 1) over the hydrological year 2007-2008 (from 17 October to 10 October). Black curves represent the simulated mass balance with corrected forcing (section 2.3.4). Pink curves are the simulations without the incoming longwave radiation correction. Blue curves are the simulations without the wind speed correction. Black dots represent the measurements and their uncertainties.

**4.2.2.3 Sensitivity to wind speed correction**

The impact of wind speed on the simulated mass balance was assessed over the period 2006-2010 using the wind speed data from $AWS_m$ and from SAFRAN (Figure 7, black and blue curves). The mean difference at the end of the hydrological year, considering all stakes, is -0.70 m w.e. $yr^{-1}$ (with a standard deviation equal to ±0.76 m w.e. $yr^{-1}$), with a maximum difference of -1.72 m w.e. $yr^{-1}$ (Stake #16 in Figure 7). The use of uncorrected wind speed data significantly decreases the performance of the annual SMB simulations (the NS coefficient decreases from 0.67 to -0.04 (Table 4)).

The influence of wind speed and direction on snow accumulation variability during and after snowfall events is widely recognized (e.g., Winstral and Marks, 2002). Our results emphasize the important role of wind speed in energy balance exchanges and its impact on the SMB (Figure 7). Indeed, wind impacts the snow surface density through snow compaction (e.g., Vionnet *et al.*, 2012) and the turbulent fluxes (e.g., Litt *et al.*, 2017). Actually each component of the turbulent fluxes (H and LE) simulated with original SAFRAN wind data is lower than those simulated with the measured wind. For instance, the mean value of H computed over summer 2006 is equal to 7.2 W $m^{-2}$ (with a standard deviation of 10.7 W $m^{-2}$) when simulated with SAFRAN wind data, compared to 22.2 W $m^{-2}$ (with a standard deviation of 37.8 W $m^{-2}$) when using measured wind speed.

Considering wind speed data from $AWS_m$ leads to an increase in the snow density of about 50 kg $m^{-3}$ for the upper layers of the snowpack when density is lower than 300 kg $m^{-3}$. Above this value, densities are similar with and without wind speed correction. The changes in snow density directly affect the thermal conductivity of the upper snow layers (Yen, 1981).

As a consequence, differences in snow density, and even more so in turbulent fluxes, due to wind speed correction have a considerable impact on surface temperature (Figure 8). Over the period 2006-2010, the mean simulated snow surface temperature increases by 3.4 °C using corrected wind speed (maximum increase of 20 °C) and the mean ice surface temperature increases by 2.7 °C (maximum increase of 10 °C), with larger differences during the night.

Simulated surface temperatures were compared to measurements. During winter, when snow depth is sufficient (~20 cm) and energy balance not affected by ground fluxes, outgoing LW measured at $AWS_m$ can provide the snow surface temperature using the Stephan Boltzmann law (with an uncertainty of 1°C). During summer, outgoing LW measurements from $AWS_{g06}$ were used to compare simulated and measured surface temperatures. Figure 8 illustrates the impact of the wind correction on the simulated surface temperature and the comparison with measurements in 2006. Results indicated a significant increase in correlation between measured and simulated surface temperatures when corrected wind was considered (NS increase from -3.14 to 0.28 for the summer period and from -0.30 to 0.20 for the winter period). Nevertheless, even using corrected wind speed values, simulated surface temperatures are still lower than the measurements, especially over the winter period. Note that the surface temperature also has a feedback on the turbulent fluxes (e.g. an increase in ice/snow surface temperature can reduce the turbulent heat flux into the surface), leading to a complex relation between these variables.

During the winter season, surface temperatures (measured or simulated) are in any case too low for melting to occur and consequently the impact of the correction of wind speed on the winter SMB is negligible (Figure 7). The impact of surface temperature can be first observed in spring: if surface temperature during the night is too low, the available energy during the day is used only to warm the snow layer and not for melting. A larger impact of the correction of wind on SMB can be observed during the second part of the summer in the ablation area (from about July, Figure 7) when the surface is ice, indicating the importance of having wind speed measurements to compute turbulent fluxes.

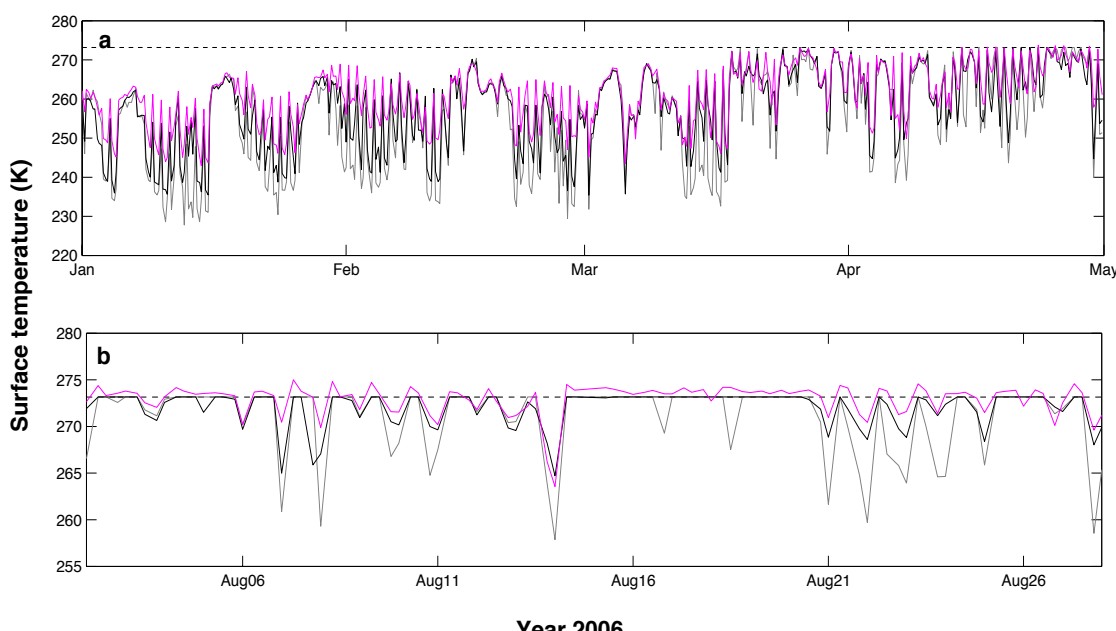

**Figure 8.** (a) Winter hourly surface temperature (snow surface) from January to May 2006, measured using outgoing LW radiation at $AWS_m$ (pink) and simulated without (grey) and with (black) wind speed correction. (b) Summer hourly surface temperature measured using LWout at $AWS_{g06}$ (pink) and simulated without (grey) and with (black) wind speed correction from 9 July to 28 August 2006. Dashed line corresponds to the melting point.

### 4.2.3 Sensitivity of Crocus parameters

As mentioned in the previous section, even when considering measured wind speed, a difference persists between the measured and simulated surface temperatures. Sensitivity tests were performed to better understand the processes responsible for this under-estimation of simulated surface temperatures.

#### 4.2.3.1 Surface roughness length

While feedback loops exist between turbulent fluxes and surface temperature, we attempted to assess the impact of effective

roughness length values on both surface temperature and summer SMB Considering that the values generally found in the literature for ice surfaces are most of the time in the range of 1 to 6 mm (e.g. Brock *et al.,* 2006, Table 1; Smith *et al.,* 2016) and can reach 80 mm for very rough glacier ice (Smeets *et al.,* 1999), the tests were performed with ice values ranging between 1 and 100 mm (Figure 9). Figure 9a illustrates a stronger impact for more negative SSBMs (corresponding to mainly ice ablation) than for the less negative summer SMBs (corresponding to snow ablation). This is confirmed by results shown in Figures 9b (one stake in the ablation zone) and 9c (one stake in the accumulation area). At the end of the hydrological season, there is a difference of SMB of 77% in the ablation area (ice surface) and 11% in the accumulation area (snow surface) between roughness values of 1 and 100. In fact, changing the roughness length considerably affects the simulated ice ablation (Figure 9b) but the effect is limited on the simulated snow ablation (Figure 9c), considering that the impact of other parameters such as albedo changes can be greater

In this study, $z_0$ is calibrated to provide good agreement between the simulated and measured turbulent fluxes on the ice from 9 July to 28 August 2006 (Litt *et al.,* 2017; see section 2.3.2). For this, numerical experiments were performed using an ice roughness length $z_{0ice}$ ranging from $10^{-5}$ to 0.2 m. The best simulation performed with Crocus was obtained with an ice roughness length ($z_{0ice}$) of 1 mm. Note that $z_0$ was calibrated by fitting the simulated sum of H and LE with the one calculated with the EC method. However, turbulent flux measurements are available over a short time period, only for one ablation season. As $z_0$ can vary considerably over time (including at daily timescales) and space (snow or ice surfaces), and due to the strong sensitivity of the model to this parameter and the large uncertainty in its determination, having *in-situ* turbulent flux measurements over ice and snow surfaces, and covering various summer ablation seasons is very useful to properly calibrate $z_0$. In addition, uncertainty evaluation should be further investigated, for instance, by testing the surface renewal method to improve the scalar roughnesses estimation, or in considering stability corrections (Andreas, 1987).

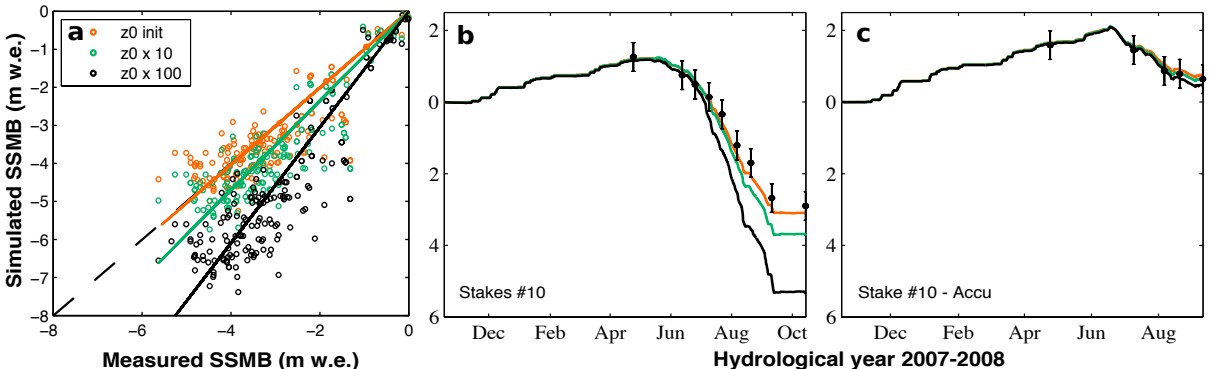

**Figure 9.** (a) Correlations between simulated and measured summer SMBs at each measurement point over the 2006-2010 period, using different roughness lengths. Surface mass balance evolution at one stake in ablation area (b) and at one stake in accumulation area (c) over the hydrological year 2007-2008. SMB was simulated using different roughness length values for snow and ice ($z_{0ice}$ = 1 mm and $z_{0snow}$ = 0.1 mm (orange); $z_{0ice}$ = 10 mm and $z_{0snow}$ = 1 mm (green) and $z_{0ice}$ = 100 mm and $z_{0snow}$ = 10 mm (black)).

**4.2.3.2 Ice albedo**

Albedo measurements were used to calibrate and validate the albedo range in the model. The correlation between daily albedo measurements and daily simulations is significant (95% confidence level (Student's t test), $R^2 = 0.54$, RMSE = 0.23). In particular, the transition date from snow surface to ice surface is well represented in the ablation area (difference lower than 5 days). Nevertheless, surface albedo is highly variable in time and space and validation was carried out at only two points (one in the ablation area and one in the accumulation area) over a short time period (three months). Therefore, this

variable was tested to evaluate its sensitivity.

Simulations to test Crocus parameter sensitivity to ice albedo were performed in the ablation area for the 2006-2011 period by changing the ice albedo of the spectral band [0.3-0.8] μm from 0.16 to 0.32. For the sake of clarity, Figure 10 illustrates the results for stake #10 for the hydrological year 2007-2008, but results are similar for the other stakes and over the other hydrological years. The difference in annual SMBs at the end of the hydrological year is 0.48 m w.e. $\mathrm{yr}^{-1}$ (17% of the annual

SMBs). Our results point out that this parameter needs to be properly calibrated with measurements on Saint-Sorlin glacier to optimize model performance. However, model performance is possibly more sensitive to albedo parameterization for glaciers where radiation has a larger contribution to melt energy. Six *et al.* (2009, Table 3) showed that, during summer 2006, monthly mean fluxes of the energy balance were 80% for the net all-wave radiation R and 20% for the turbulent fluxes. This monthly mean distribution can reach 70% and 30% respectively on a daily time scale. For Zongo Glacier

(Bolivia, 16° S) the net all wave radiation R can represent 97% of the energy balance (Sicart *et al.,* 2008).).

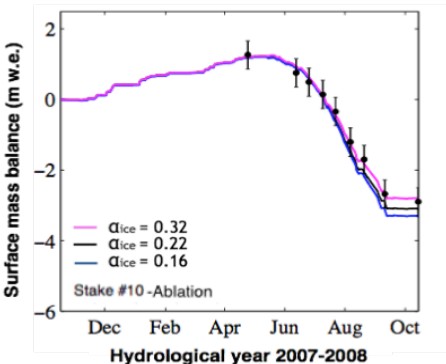

**Figure 10.** Surface mass balance evolution with an ice albedo calibrated at 0.16 (blue), 0.22 (black) and 0.32 (pink), at one stake in ablation area over the hydrological year 2007-2008.

**4.2.3.3 Liquid water content at the surface**

During melting events the simulated liquid water percolates through the snow layers when the liquid water volumetric content exceeds 5% of the pore volume (Vionnet *et al.*, 2012). For ice, the porosity is set to 0 so the liquid water

immediately flows off the glacier and cannot remain at the surface. The use of such parameterization on ice is questionable as water from ice melt or from shallow snow layer melt above ice can stay at the surface.

A sensitivity test was performed, considering ice as a porous material, able to store liquid water in 10% of its total volume. Note that this sensitivity test is an over-simplistic way to consider the presence or not of water at the ice surface. As the water percolates very quickly, the test can be performed over a very short time period during summer (few days). For summer 2006 (not shown), a significant difference in the simulations was found for the surface temperature (maximum of 20 °C difference) and surface mass balance (difference of 0.6 m w.e. after 15 days of simulation) when the possibility for water to be stored at the glacier surface was considered or not. While such test is simplistic, it indicates the significant sensitivity of the energy budget to the presence of liquid water on the ice surface. This process deserves to be properly taken into account in Crocus when using the snow model over ice surfaces, as has been done for summer SMB simulations in Greenland with other models (e.g., Gallée and Duynkerke, 1997; Lefebre, 2003; Fettweis, 2007).

## 5. Conclusion

This study has evaluated the performance of the Crocus snowpack model, which was fed with SAFRAN reanalysis data, thereby simulating seasonal and annual SMBs of Saint-Sorlin Glacier over the last 20 years. Using meteorological forcing adjusted with *in-situ* measurements, our results show very good performance of the model to simulate summer SMB in both accumulation and ablation areas. Performance of the model is lower for the 1996-2005 period due to the absence of *in-situ* meteorological measurements to adjust the forcing data.

Additionally, this study compared the performance of this energy balance model to an empirical approach using temperature and potential incoming solar radiation as inputs. Regarding simulations of summer SMB for the accumulation area, our results show better performance using the energy balance model, especially concerning simulations of snow and firn melting in the accumulation area. Regarding the ablation area of the glacier, the two approaches show similar performance when forced with meteorological data adjusted with nearby AWS measurements. When such measurements are not available in the vicinity of the glacier, performance of the empirical model in the ablation area is superior although the physical processes are not properly represented. However, the temporal stability of the calibration parameters of the empirical approach need to be assessed over a longer time period before using such an approach over several decades.

These conflicting conclusions about model performance in accumulation and ablation area emphasize greater importance of having meteorological data to correct the forcing in ablation area. Indeed, in our sensitivity study using forcing data, the results demonstrate that the Crocus model is highly sensitive to wind speed, especially for ice melt simulations. Indeed, using *in-situ* wind speed data instead of reanalysis data (where observed wind speed values larger than 10 m s$^{-1}$ can be under-estimated by a factor 2 or 3) led to an annual mass balance decrease of more than 1.7 m w.e. yr$^{-1}$. Thus, without local wind speed measurements, the model's performance strongly decreases, even using wind speed data corrected via a quantile-mapping method. In addition this study confirms the findings by Dumont *et al.* (2012) concerning the importance of

correcting the incoming longwave radiation from SAFRAN.

Model calibration represents an important step to improving model performance. According to the sensitivity study concerning model calibration, our results highlight the importance of calibrating the ice surface roughness using turbulent fluxes measurements. An increase in $z0_{ice}$ by a factor of 10 can have an impact of 1.5 m w.e. yr$^{-1}$ on ice melting. Regarding the ice albedo, while having *in-situ* measurements to calibrate the model improved model performance; the sensitivity of summer SMB for this variable is lower than the sensitivity to wind speed over icy surfaces (the ice melt difference reaches 0.48 m w.e yr$^{-1}$ when the ice albedo is divided by a factor 2). This could suggest a relatively low sensitivity to ice albedo change (due to dust or black carbon for example) for summer SMB variations in the future."

While both these approaches can provide good summer SMB simulations, winter SMB simulations need to be corrected using winter mass balance measurements. In any case, our results indicate a strong sensitivity of annual SMB to winter SMB. The understanding of the spatio-temporal variability of accumulation processes at the glacier surface needs to be more fully investigated in future work.

In conclusion, our study reveals the major role of wind speed, which controls the magnitude of turbulent fluxes, on melting. The results highlight a very serious obstacle for the modelling of future glacier mass balances, as this meteorological variable is highly unpredictable. Our results also suggest that the sensitivity of annual mass balance to accumulation and wind speed parameters is of primary significance, as compared to the sensitivity to snow and ice albedo changes. However, as such data are still difficult to represent in climatic models, the accuracy of their predictions are also questionable (e.g. Terzago *et al.*, 2017). We thus suggest a careful use of the physical approach for future long-term simulations, considering the uncertainties. Nevertheless, despite these limitations for future simulations, this physical model remains crucial to study and understand physical processes and interactions between atmospheric variables and ablation. Otherwise, although empirical approaches based on simple meteorological variables also have serious drawbacks, they could be more appropriate for simulations of glaciers in the future, especially to simulate summer SMB in ablation areas, bearing in mind the lack of availability of reliable information on future meteorological variables and surface roughness.

**Acknowledgements.** This study was conducted in the context of the French *Service d'Observation* GLACIOCLIM (https://glacioclim.osug.fr). We would like to thank everyone who helped in collecting data during these glacier field campaigns. Most of the computations presented in this paper were performed using the Froggy platform of the CIMENT infrastructure (https://ciment.ujf-grenoble.fr), which is supported by the Rhône-Alpes region (grant CPER07_13 CIRA). The work was made possible by the contributions of Labex OSUG@2020 (Investissements d'avenir – ANR10 LABX56), the ANR program TAG 05-JCJC-0135 (conducted by J.E. Sicart) and the French Research Ministry. We thank Y. Durand and G. Giraud (CNRM-GAME/CEN) for providing the SAFRAN data. Finally we acknowledge the editor and the two anonymous reviewers for detailed comments and helpful suggestions on previous versions of the manuscript.

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
