# Peer review of "Relative performance of empirical and physical models in assessing the seasonal and annual glacier surface mass balance of Saint Sorlin glacier (French Alps)"

_The Cryosphere, 2017_

## Referee Comment (RC1) · Anonymous Referee #1 · 27 Oct 2017

This study evaluates the performance of the Crocus snowpack model and a temperature index model in simulating seasonal and annual surface mass balance of the Saint-Sorlin Glacier in the French Alps. The models are forced using SAFRAN reanalysis data; both in raw form, and corrected using in situ AWS measurements. The authors also examine model performance sensitivity to season, DEM resolution and temporal variation, inputted wind speed, albedo, and roughness lengths. The authors conclude that model performance decreases substantially without in situ meteorological measurements, and that without access to these measurements, an empirical temperature index model may be more appropriate than using an energy balance approach for future mass balance projections. The authors also state that Crocus model performance

was more sensitive to wind speed input than ice albedo.

This study presents a useful exercise in evaluating multi-year modelling of surface mass balance and highlights some of the uncertainties in the methods. It will make a good addition to the literature, but I recommend some changes before publication; outlined in detail below. Broadly, the paper suffers from a lack of detail, both in describing some of the methodology, and in presenting only limited portions of the observations and results. As a result, some of the findings and conclusions of the paper feel unsupported. The choice of metrics used to present results, in some cases, limits the information provided to the reader, and may be inappropriately used in places. Some sources of uncertainty are not mentioned in the study, and while investigating each of these may be beyond the scope of the paper, they should at least be recognised.

Specific Comments: P1L26: 'an energy balance model'.

P1L27: 'requires'.

P1L27 – 29: 'With the current...' This sentence is unclear in its meaning. Consider rephrasing to clarify you are referring to the uncertainty in the temporal evolution of the relevant meteorological and surface properties of individual glaciers.

P2L28 – 31: Include references for both energy balance approaches mentioned here (parameterised and complete components).

P3L21: Include a few more details that are relevant to this study, for example slope angle and aspect at each station location. These could also be added to table 1.

P3L28: Missing an m in units for $\pm 0.15$ m w.e. yr-1.

P3L29: Include units for 0.30.

P4Figure1: Remove black outline from triangles in legend or add them to those used on the map.

P4L10 – 12: Provide some brief details on the DEM creation and kriging method, and

references if these maps were created in a previous study.

P4L18: Was the relative humidity sensor also in the aspirated shield? I'm assuming T and RH are from the same sensor (HMP45C), so clarify this here.

P4L18: Specify which directions (incoming, reflected/outgoing) you have measured for short and longwave radiation (as given in table 1).

P4L19: 'Data were quality. . .' This statement is very vague. Please provide detail on the quality control applied.

P5L1 – 4: Are the glacier stations mast mounted or sitting on the ice surface (relevant for maintaining constant height above surface)? Please specify. A picture of the stations would be useful. The height of the wind sensor on AWSm is mentioned in section 2.3.4 as being around 2 m. Can you provide specific height values for this and the wind sensors on the glacier stations? Near surface gradients of wind speed can be substantial.

P5L10: Some further details relevant to this study would be useful (in addition to supplementary details in the referenced paper). For example, the installed height of the EC system and how much on average its height above the surface ranged in between adjustments.

P5L24 – 25: No further details on the slope correction are provided in section 3.1.1. Some additional information would be useful, such as the slope angles at each station, how the partitioning of diffuse and direct incoming shortwave is estimated in SAFRAN (could this be affected by the presence of low-altitude clouds not considered by SAFRAN, as mentioned in section 2.3.4?), and what the impact of the slope correction is on the magnitude of the shortwave fluxes at the station sites. These values could also be useful when discussing the effects of changing the DEM resolution in section 4.2.1.

P6Table1: Variables and instruments are not properly aligned in table.

P7L5: Section 4.2 rather than 4.3

P7L6 – 8: In addition to R2 values, the mean bias error (or similar metric) between the SAFRAN and AWS-observed variables would be a useful metric to include to quantify any bias or lack off.

P7L8 – 10. 'The comparison between SAFRAN and AWSm...' I may be misinterpreting you here, but I'd like to clarify what your meaning is. You are saying that when there is little cloud cover, SAFRAN overestimates incoming longwave. Why then would the presence of low-altitude cloud, which has not been considered by SAFRAN be an explanation for this? Surely, the low-altitude cloud would lead to an increase in observed incoming longwave relative to the SAFRAN data?

P7L17: How do the estimates of incoming (slope-corrected) shortwave from SAFRAN compare with the values you calculate from the observed reflected shortwave and the albedo scheme?

P7L19: What is meant by wind speed 'generally considered' to be at 2 m?

P7L21 – 22: 'This underestimation is ...' This statement could use a reference to supporting studies. In addition, it would be useful to present a comparison of wind direction in addition to wind speed. Is the observed wind direction in the downslope direction during these periods of large wind speed differences and suspected katabatic flow? How well does SAFRAN represent the local wind direction i.e. is the influence of the glacier accounted for?

P9L4 – 5: Comma required between 'model, implemented' and '2013), was'.

P9L9 – 10: How are the empirical values selected for microstructure? Are they glacier specific? What is the uncertainty in them?

P9L17: The density value of 917 kg m-3 is generally assigned to pure glacier ice. Have you examined your model's sensitivity to varying this value i.e. to account for uncertainty in the actual ice density?
P10L10: When running the ATI, what data are used for the incoming potential direct solar radiation (IPOT)? Is this simply the incoming direct shortwave radiation from SAFRAN?

P10L16: I know the ATI model uses the WSMB from crocus, but is summertime precipitation and potential summertime snowfall accounted for when using the ATI?

P10L31 – 32: 'Peformance was evaluated...' Strictly speaking, you are not evaluating ATI performance with winter SMB measurements as you are using Crocus WSMB values (as mention above in L26 -27). Consider restructuring this sentence.

P11L33: Typo; missing an 'in' between presented and section, and section number should be 4.2.3.

P12L22 – 24: 'Indeed, the ATI...' This sentence needs to be rewritten, and the figure reference should be Figure 4a. Looking at Figure 4a, when observed SSMB is above -2 m w.e. (i.e. less negative than -2), the ATI model shows a tendency to underestimate SSMB (more negative), or in other words, to overestimate ablation.

P12L26 - 28: 'In addition, the temporal evolution...' By temporal evolution, do you mean you compared the cumulative simulated surface mass balances from the two models over the summer? If so, was this using a daily time step? Was this carried out just for one summer season or all? You have presented the maximum differences for the SSMBs, but perhaps if you are interested in the temporal evolution, it would be more interesting to describe the biases/differences that occurred within the season (e.g. are the SSMB differences driven by a general bias or by individual days with large differences etc).

P12L30: 'Here again...' Following on from my point above for L22 – 24, SSMBs in the accumulation area from the ATI model appear to be underestimated; it is the ablation that is overestimated.

P13Figure3: Axis labels should specify surface mass balance (SMB) not MB.

[Figure]

P14Figure4: Same point as for fig 3. Also, in caption, using 'a-c' etc. suggests a to c, inclusive (i.e. a,b,c). Consider using 'a and c'.

P14L8 – 10: 'various stakes'. How many stakes was this performed on? Some results/plots from other stakes would be useful. Maybe provide the mean and standard deviation (over all tested stakes) of the differences in annual SMBs for the extreme summers and winters mentioned here.

P15L5: Have you quantified this as 'significant'?

P15L23: 'First, correlations were computed. . .' Correlations with what?

P17L1 – 4: Repeated referencing of table 3 is probably unnecessary in the same subsection.

P17Table4: Caption appears to be the same as that used for table 3. Why are the performances for ASMB without LW and wind corrections so poor compared to the seasonal performances?

P18Figure6: It would be interesting in these plots to see the effect of removing the precipitation correction also.

P18L11: Swap the position of 'blue' and 'black' to avoid confusion.

P18L13: 'The use of. . .' Ensure that you refer to correcting the wind speed data rather than 'wind data', as you have not discussed correcting the wind direction in this paper. The 'Without wind correction' label in Table 4 should also be corrected.

P19L5: Provide mean values of the turbulent fluxes using observed and SAFRAN wind speeds.

P19L11 – 12: Replace 'up to' (e.g. 'up to 20$°$C') with 'max. increase of'. The use of 'up to' here sounds like the surface temperature is increased to this temperature!

P19L21: Can you clarify what you mean by a positive feedback in this case? An

increase in ice/snow surface temperature can reduce the turbulent heat flux into the surface.

P19L23 – 24: Add speed to '…correction of wind on the winter…' Also, just to clarify, does wind speed not affect how the model deals with snow accumulation?

P19L26 – 27: 'A larger impact…' Are you suggesting this is due to it being an ice surface or because the surface is warmer at this time and melt is occurring?

P20Figure7: Explain dashed line (i.e. melting point).

P20L8 – 9: Why only test for roughness values that were larger than the employed roughness value?

P20L10 – 13: It would be useful to present the values for the % difference in SMB between the different roughness value scenarios, as it is hard to distinguish in figure 8 if there really are different responses for snow and ice surfaces, or if it just that the magnitudes are greater because there is greater levels of ablation over the ice surfaces. Clarify what you think your results are showing. Are you suggesting that varying the implemented roughness lengths for snow will not affect the turbulent heat fluxes estimated by the model over this surface? This would not follow the theory of the bulk aerodynamic method implemented in the model.

P20L17 – 18: 'Note that…' Was this a single sum over the full season, or did you examine the simulated fluxes over shorter timescales e.g. daily sums? Using shorter timescales might allow you to look for the temporal variation in roughness length you have mentioned.

P20L20: You have arbitrarily selected the values for the scalar roughness lengths to be 1/10th of that for momentum. However, there are other schemes suggested in the literature to estimate the scalar values, such as assuming a 1/100 ratio, assuming they are all equal, and utilising a surface renewal method (Andreas, 1987). Have you considered the model sensitivity to these values? Another major source of uncertainty

in the estimating the turbulent fluxes is how the model accounts for changes in atmospheric stability. While an investigation of this uncertainty may be beyond the scope of this paper, it is worth mentioning it as a source of uncertainty, particularly when using reanalysis data where the localised effects of the glacier on stability may not be resolved. Andreas, E. L. (1987). A theory for the scalar roughness and the scalar transfer coefficients over snow and sea ice. Boundary-Layer Meteorol. 38, 159–184. doi: 10.1007/BF00121562

P21Figure8: Reposition '(b)' and '(c)' in the caption to be in front of the relevant text.

P21L14: This finding regarding the importance of ice albedo to model performance may not be very transferable. The same test performed on a glacier in a different climate setting (e.g. one where radiation has a larger contribution to melt energy) may find model performance to be more sensitive to albedo parameterisation.

P22L31 – P23L2: Following on from the point above, it would be interesting to see the partitioning of the melt energy for the study glacier, and a summary of the meteorological conditions. These model sensitivity findings may be very dependent on the ratio between the turbulent heat and radiation fluxes.

---

## Referee Comment (RC2) · Anonymous Referee #2 · 8 Nov 2017

Review of Relative performance of empirical and physical models in assessing seasonal and annual glacier surface mass balance in the French Alps by Reveillet et al

The paper by Reveillet and others describe the results of the modeling of the surface mass balance of Saint-Sorlin glacier in the French Alps. The SMB is calculated with CROCUS, an energy balance model originally designed to calculate snow pack. This model is forced with SAFRAN reanalysis data that is corrected to the in-situ weather observation at the location of the glacier. The authors explore the sensitivity of the simulated SMB to different components of the climatic forcing and to various model parameters. Furthermore is the result of the energy balance model compared with the

results of an empirical model. The study is well written and the results are interesting. I have the following comments, spit up in general comments, specific points and some technical points, the latter two per page and line number.

general points

- the difference between energy balance model and empirical model is not just the model formulation, but also the way the models are calibrated. The parameters of the empirical model are calibrated by fitting calculated SMB to the observed SMB, while for the energy balance the meteorological input of the model is fitted to observed weather. Would the energy balance model perform better if its model parameters and input corrections are optimized to the observed SMB?

- I do not fully see how the conclusion that empirical models would be better suited to model glacier SMB for periods, or glaciers, without AWS measurements. The results of this paper show that the energy balance model performs better, mainly in the accumulation area, than the empirical model. This is also true for the period where the input of the model is not directly corrected with the AWS data. So therefore I do not see why the empirical model should be more suited to model future glacier mass balance, when also no direct correction of the meteorological forcing with in-situ observation is possible.

- It would be instructive include information and figures on the climate at Saint-Sorlin Glacier in the paper. And to include in these figures a comparison between the original climate given by the SAFRAN data and the climate as measured with the AWS.

specific points

page 2: -l7 in addition to the fact that it is questionable if the calibrated parameters are valid over long time periods, a disadvantage of the temperature-index models is that the parameters have no validity for other glaciers. Maybe you could add this here.

page 3 - l2-5 Here you discuss variability, but you are, also, interested in the absolute

value of the SMB, not only of the variations from year to year. - l20 I do not see that the aspects of Saint-Sorlin vary a lot in Figure 1, it is mainly N - NE - E slopes. Please be a bit more specific.

page 4 - Figure 1 Could you add to surrounding topography in Figure 1b? That could give a better understanding for your discussion of wind speed and snow distribution. - l10 What is the accuracy of the DEMs? Do they give the same values for stable ground?

page 5 - l23 So SAFRAN is not gridded, but has one output per mountain region as given in Figure 1a? And this is then distributed following elevation and aspect?

page 7 - l5 High correlation doe not say it all, as you can have a significant bias while having a high correlation. Could you indicate how well the values correspond, for example with a RMSE? - l9-10 How does a lack of low-altitude clouds in SAFRAN explain an overestimate of incoming long-wave radiation in SAFRAN? I would expect that low altitude clouds are warmer than high-altitude clouds and thus emit more long wave radiation, such that not including these low-altitude clouds would lead to an underestimate of the incoming long wave radiation. - l17 Also here, could you give an estimate of the bias/rmse in addition to the correlation? - l22 How likely is it that you measured the katabatic wind on the site of AWSm? It is located off the glacier and well above the nearest glacier surface (glacier extends to below 2700 m and the AWSm is at 2720 m), while the katabatic wind on a small glacier can be quite shallow. If you look at the wind direction, do you then see a consistent down-slope wind?

page 10 - l30 It is not clear why you perform the calculations on a 200 m resolution. Here you refer to 2.2.2, where in turn is referred to 2.3.3 where it is stated 'we linearly interpolated on the 200-m horizontal resolution grid' without any further reason why this interpolated to this 200 m. Please explain.

page 11 - l10-33 I would add the content of these section 3.2.2.x to the respective subsections in the results section 4.

page 12 - l10-11 Here you write that the interpolation used to create the precipitation can explain the difference between measured and simulated winter SMB. But if all winter SMB measurements are included in the determination of the precipitation fraction maps, then how does the interpolation between these points affect the model results on the stake locations? And how do melting events in the accumulation period explain the difference? They should be in the simulations as well? - ln29 This contradicts the conclusion that when no correction in the forcing is possible due to lack of AWS data, the ATI performs better than the energy balance simulations

page 14 - Figure 4 You do not plot the correlation here. - section 4.1.3 I think the results of this section could be clarified with a scatter plot where you plot annual SMB vs summer SMB with constant winter SMB and one plot with annual SMB vs winter SMB for constant summer SMB.

page 16 - l6-9 This part is not so clear to me. How can differences up to 25% be explained by 'only slightly affect the simulated SMB for a limited number of stakes'?

page 23 - l24 I do not really understand the conclusion that empirical models would be better fitted to model future SMB. From your results it is clear that using observations to correct forcing the energy balance model performs better than the empirical model, also for the period where no AWS is available. It is unclear whether the forcing corrections remain valid in the future, but the same holds for the parameters in the empirical model (as you have pointed out). So why is then the empirical model more reliable for future projections?

technical points

page 1 - Why not include Saint-Sorlin in the title, replacing the French Alps?

page 3 - l23 Here and elsewhere in the paper: I am not a big fan of these acronyms WSMB, ASMB, SSMB. I feel it provides easier reading by just writing out 'winter SMB', 'annual SMB', and 'summer SMB'. - Table1: add some separation between the different

stations, like a line, or a blank text line.

page 5 - l10 Please rephrase this sentence on the instrumentation height adjustment. You probably mean that you lowered the instruments in order to keep the distance between instruments and the ice surface constant. - l27 'emitted long wave radiation and reflected short wave radiation', the earth surface does not emit short wave radiation.

page 7 -l9 could you replace 'explained' with 'caused'? 'explained' would require a more in-depth analysis

page 9 - l24 If I understand correctly, you have changed the lower albedo limit from 0.7 to 0.5, keeping the time decay. Then I suggest to replace 'fixed at' with 'set to', as 'fixed' could indicate that you eliminated the time evolution of the albedo.

page 14 - Figure 4 caption: replace the hyphen in 'blue (a-c)', 'orange (b-d)', etc. with a comma 'blue (a,c)'

---

## Author Response (AR1)

**Response to the anonymous referee #1**

**General comment:**

*This study evaluates the performance of the Crocus snowpack model and a temperature index model in simulating seasonal and annual surface mass balance of the Saint-Sorlin Glacier in the French Alps. The models are forced using SAFRAN reanalysis data; both in raw form, and corrected using in situ AWS measurements. The authors also examine model performance sensitivity to season, DEM resolution and temporal variation, inputted wind speed, albedo, and roughness lengths. The authors conclude that model performance decreases substantially without in situ meteorological measurements, and that without access to these measurements, an empirical temperature index model may be more appropriate than using an energy balance approach for future mass balance projections. The authors also state that Crocus model performance was more sensitive to wind speed input than ice albedo.*

*This study presents a useful exercise in evaluating multi-year modelling of surface mass balance and highlights some of the uncertainties in the methods. It will make a good addition to the literature, but I recommend some changes before publication; outlined in detail below. Broadly, the paper suffers from a lack of detail, both in describing some of the methodology, and in presenting only limited portions of the observations and results. As a result, some of the findings and conclusions of the paper feel unsupported. The choice of metrics used to present results, in some cases, limits the information provided to the reader, and may be inappropriately used in places. Some sources of uncertainty are not mentioned in the study, and while investigating each of these may be beyond the scope of the paper, they should at least be recognised.*

**Authors' response:**

We thank the reviewer for this constructive and thorough review of the manuscript.

As requested, some of the methodology has been described in more detail where necessary (see for example the answers to comments n°5, 9 and 13 below). More results are presented (e.g., specific comments n°32 or 56), the metrics have been extended (comment n°18) and some sources of uncertainties have been more fully described (in the discussion section) or at least mentioned if their investigation is beyond the scope of the paper.

Please find more details about your comments below. Note that replies from the authors are in green.

**Specific comment #1 (P1 L26):**

*'an energy balance model'*

**Authors' response:**

'an' has been added

**Specific comment #2 (P1 L27):**

*'requires'*

**Authors' response:**

'required' has been replaced by 'requires'.

**Specific comment #3 (P1 L27-29):**

*'With the current…' This sentence is unclear in its meaning. Consider rephrasing to clarify you are referring to the uncertainty in the temporal evolution of the relevant meteorological and surface properties of individual glaciers.*

**Authors' response:**

The sentence has been changed in response to your comment: 'Given the uncertainties in the temporal

evolution of the relevant meteorological variables and glacier surface properties in the future, empirical approaches based on temperature and precipitation could be more appropriate for simulations of glaciers in the future.'

**Specific comment #4 (P2 L28-31):**
*Include references for both energy balance approaches mentioned here (parameterised and complete components).*
**Authors' response:**
References have been added both for parameterized and complete energy balance approaches:
"Due to the lack of measurements and the complexity of measuring each of the components of the energy balance (especially turbulent fluxes), physically based models are generally calibrated by adjusting certain parameters (e.g., roughness length to quantify turbulent fluxes) to fit with SMB measurements balance (e.g. Dumont et al., 2012)".

**Specific comment #5 (P3 L21):**
*Include a few more details that are relevant to this study, for example slope angle and aspect at each station location. These could also be added to table 1.*
**Authors' response:**
As slope angle and aspect are similar for all stations located on the glacier, the following information has been added in the caption of Table 1:
"Slope angles and aspect are similar for all stations located on the glacier (AWSg06, AWSg-accu08, AWSg08, AWSg09), respectively 5° and roughly North. Slope at AWSm is 0°".

**Specific comment #6 (P3 L28):**
*Missing an m in units for 0.15 m w.e. yr-1.*
**Authors' response:**
'm' has been added

**Specific comment #7 (P3 L29):**
*Include units for 0.30.*
**Authors' response:**
Done.

**Specific comment #8 (P4 Figure1):**
*Remove black outline from triangles in legend or add them to those used on the map.*
**Authors' response:**
Black outline has been removed from triangles in the legend.

**Specific comment #9 (P4 L10-12):**
*Provide some brief details on the DEM creation and kriging method, and references if these maps were created in a previous study.*
**Authors' response:**
These maps were created for this study. In response to your comment, the following information has been added:
"These DEMs were derived from aerial photogrammetry and have a 10-m spatial resolution. For consistency with the resolution of the atmospheric data described in section 2.3.3, they were, for this study, upscaled to 200-m resolution using the kriging method (based on the default linear variogram) of SURFER mapping software (Golden Software, LLC)."

**Specific comment #10 (P4 L18):**

*Was the relative humidity sensor also in the aspirated shield? I'm assuming T and RH are from the same sensor (HMP45C), so clarify this here.*

**Authors' response:**

The following sentence has been changed in response to your comment (as well as comment #11):

"This AWS records 2-m air temperature and relative humidity (the common sensor is housed in a mechanically aspirated shield), incoming and reflected short-wave radiation, incoming and outgoing long-wave radiation, and wind speed and direction with a half-hour time step."

**Specific comment #11 (P4 L18):**

*Specify which directions (incoming, reflected/outgoing) you have measured for short and longwave radiation (as given in table 1).*

**Authors' response:**

The following sentence has been changed in response to your comment (as well as comment #10):

'This AWS records 2-m air temperature and relative humidity (the common sensor is housed in a mechanically aspirated shield), incoming and reflected short-wave radiation, incoming and outgoing long-wave radiation, and wind speed and direction with a half-hour time step.'

**Specific comment #12 (P4 L19):**

*'Data were quality…' This statement is very vague. Please provide detail on the quality control applied.*

**Authors' response:**

Right, details have been added:

"$AWS_m$ data were quality checked to avoid any problem related to a sensor malfunction: missing data were detected and reported, unrealistic values were removed and the series was compared with series from *Meteo France* network stations in the valley to identify potential bias."

**Specific comment #13 (P5 L1-4):**

*Are the glacier stations mast mounted or sitting on the ice surface (relevant for maintaining constant height above surface)? Please specify. A picture of the stations would be useful. The height of the wind sensor on AWSm is mentioned in section 2.3.4 as being around 2 m. Can you provide specific height values for this and the wind sensors on the glacier stations? Near surface gradients of wind speed can be substantial.*

**Authors' response:**

Yes, all glacier stations are mounted on masts inserted in the ice. This is an important point which was not sufficiently detailed in the manuscript.

As this general comment on instrument height is valid for all stations installed on the glacier, we have added the following sentences concerning this point to section 2.3.1 rather than section 2.3.2 which is specifically dedicated to the eddy covariance system:

"Details relative to the location, the dates of records and the different sensors of these AWSs are reported in Table 1. Stations on the glaciers are mounted on masts inserted in the ice. Due to ice melt, instrument heights are not constant over time. However, at each station (except for $AWS_{g08-accu}$ where melt is limited), a sonic ranger was set up and helped determine the melt over each recorded time step. The heights of the instrument were then adjusted in our simulation using the melt determined by the sonic ranger. Every 10 to 15 days, instruments were re-adjusted manually to a set height of 2 m."

**Specific comment #14 (P5 L10):**

*Some further details relevant to this study would be useful (in addition to supplementary details in the referenced paper). For example, the installed height of the EC system and how much on average its height above the surface ranged in between adjustments.*

**Authors' response:**

The following information has been added in reponse to your comment.

"The CSAT instrument was installed 2.00 m above the surface. The melt ranges roughly between 30 to 40 cm, with a maximum of 80 cm depending on the month and the time between two visits. Every 10 to 15 days, the instrument was re-adjusted manually to a set height of 2 m."

**Specific comment #15 (P5 L24- 25):**

*No further details on the slope correction are provided in section 3.1.1. Some additional information would be useful, such as the slope angles at each station, how the partitioning of diffuse and direct incoming shortwave is estimated in SAFRAN (could this be affected by the presence of low-altitude clouds not considered by SAFRAN, as mentioned in section 2.3.4?), and what the impact of the slope correction is on the magnitude of the shortwave fluxes at the station sites. These values could also be useful when discussing the effects of changing the DEM resolution in section 4.2.1.*

**Authors' response:**

- First, as mentioned in our response to comment 5, surface slope angles are very small at each AWSg (5° or less). Moreover, the radiometer sensors were not found to be significantly off horizontal during field visits and were leveled every time small deviations from horizontal were found. For this reason, slope corrections were not applied on the measured radiation.

To explain this, the following sentence has been added to section 2.3.3: "However, as the surface slope angles were small at each station and because the radiometers were not found to be far from horizontal during field visits, no slope correction was applied to the measured shortwave radiation."

- Secondly, concerning the partitioning of direct and diffuse SW in SAFRAN, it is done with the radiative transfer scheme. This is of course affected by the presence of clouds, which is not considered in SAFRAN, but the impact should be limited since the partitioning is only used for the snow/ice albedo calculation.

**Specific comment #16 (P6 Table1):**

*Variables and instruments are not properly aligned in table.*

**Authors' response:**

Done. Separating lines have also been added to make the table easier to read.

| Station | Location | Date of records | Timestep | Variables | Instrument | Manufacturer accuracy | Associated studies |
|---|---|---|---|---|---|---|---|
| AWS$_m$ | Moraine 2720 m a.s.l. | 2005-present | 30 min | Aspirated air T (°C)
Relative humidity (%)
Wind speed (m s$^{-1}$)
and direction (degrees)
Upward SW (W m$^{-2}$)
Downward LW (W m$^{-2}$) | Vaisala HMP45C
Vaisala HMP45C
Young 05103
Young 05103
Kipp and Zonen CG3
Kipp and Zonen CG3 | ±0.2°C
3%
0.3 m s$^{-1}$
±3°
0.4%
0.4% | *Six et al.* (2009)
*Sicart et al.* (2008)
*Dumont et al.* (2012) |
| AWS$_{g06}$ | Ablation area 2770 m a.s.l. | 9 July - 28 August 2006 | 30 min | Aspirated air T (°C)
Relative humidity (%)
Wind speed (m s$^{-1}$)
and direction (degrees)
Upward SW (W m$^{-2}$)
Downward LW (W m$^{-2}$)
EC measurements | Vaisala HMP45C
Vaisala HMP45C
Young 05103
Young 05103
Kipp and Zonen CG3
Kipp and Zonen CG3
Csat3 and Licor 7500 | ±0.2°C
3%
0.3 m s$^{-1}$
±3°
0.4%
0.4%
0.4% | *Dumont et al.* (2012)
*Litt et al.* (2016) |
| AWS$_{g-accu08}$ | Accumulation area 2900 m a.s.l. | 12 July - 10 September 2008 | 30 min | Aspirated air T (°C)
Relative humidity (%)
Wind speed (m s$^{-1}$)
and direction (degrees)
Upward SW (W m$^{-2}$)
Downward LW(W m$^{-2}$) | Vaisala HMP45C
Vaisala HMP45C
Young 05103
Young 05103
Kipp and Zonen CG3
Kipp and Zonen CG3 | ±0.2°C
3%
0.3 m s$^{-1}$
±3°
0.4%
0.4% | *Dumont et al.* (2012) |
| AWS$_{g08}$ | Ablation area 2770 m a.s.l. | 11 July - 2 August 2008 | 30 min | Aspirated air T (°C)
Relative humidity (%)
Wind speed (m s$^{-1}$)
and direction (degrees)
Upward SW (W m$^{-2}$)
Downward LW (W m$^{-2}$) | Vaisala HMP45C
Vaisala HMP45C
Young 05103
Young 05103
Kipp and Zonen CG3
Kipp and Zonen CG3 | ±0.2°C
3%
0.3 m s$^{-1}$
±3°
0.4%
0.4% | *Dumont et al.* (2012) |
| AWS$_{g09}$ | Ablation area 2770 m a.s.l. | 13 June - 4 September 2009 | 30 min | Aspirated air T (°C)
Relative humidity (%)
Wind speed (m s$^{-1}$)
and direction (degrees)
Upward SW (W m$^{-2}$)
Downward LW (W m$^{-2}$) Albedo | Vaisala HMP45C
Vaisala HMP45C
Young 05103
Young 05103
Kipp and Zonen CG3
Kipp and Zonen CG3 | ±0.2°C
3%
0.3 m s$^{-1}$
±3°
0.4%
0.4% | *Dumont et al.* (2012)
*Litt et al.* (2016) |

**Specific comment #17 (P7 L5):**

*Section 4.2 rather than 4.3*

**Authors' response:**

Done

**Specific comment #18 (P7 L6-8):**

*In addition to R2 values, the mean bias error (or similar metric) between the SAFRAN and AWS-observed variables would be a useful metric to include to quantify any bias or lack off.*

**Authors' response:**

Right. The RMSE has been computed and indicated in the text in response to your comment:

- "SAFRAN and AWS$_m$ hourly air temperatures over the ablation and accumulation seasons are well correlated ($R^2$ = 0.98 (summer) and 0.99 (winter), both significant at the 99% confidence level (Student's t test), and the Root-Mean-Square Errors (RMSE) are 0.7°C (summer) and 0.76°C (winter)). Hourly SAFRAN relative humidity is also in good agreement with the AWS$_m$ data ($R^2$ = 0.74, significant at the 95% confidence level and RMSE = 13.6%.)"

- "Using this correction, the correlation between AWS$_m$ incoming LW radiation and corrected LW radiation from SAFRAN increased the correlation from $R^2$ = 0.71 to $R^2$ = 0.83 and decreased the RMSE from 44.3 W m$^{-2}$ to 29.7 W m$^{-2}$."

Note that in the previous version, the correlation regarding the LW was indicated as R and not $R^2$. This mistake has been corrected in the current version.

- "Correlations between daily incoming shortwave radiation ($R^2$ = 0.81) are significant at the 99% confidence level (Student's t test) and RMSE = 77.2 W m$^{-2}$."

- "A poor correlation ($R^2$ = 0.19, RMSE = 3.8 m s$^{-1}$) between SAFRAN wind speed (considered at 2-m)…"

- "Since the correlation between the measured wind speed on the foreland and on the glacier is high ($R^2$=0.97, RMSE=1.7 m s$^{-1}$) "

**Specific comment #19 (P7 L8-10):**

*'The comparison between SAFRAN and AWSm…' I may be misinterpreting you here, but I'd like to clarify what your meaning is. You are saying that when there is little cloud cover, SAFRAN*

*overestimates incoming longwave. Why then would the presence of low-altitude cloud, which has not been considered by SAFRAN be an explanation for this? Surely, the low-altitude cloud would lead to an increase in observed incoming longwave relative to the SAFRAN data?*

**Authors' response:**

We completely agree. This was a mistake that slipped into the original manuscript. The sentence initially read:

"The comparison between SAFRAN and $AWS_m$ incoming long wave radiation indicates an overestimation of SAFRAN data for **low** cloudiness conditions. This can be explained by local orographic features and/or **low**-altitude clouds that are not considered in SAFRAN reanalysis."

It should be, in reality: "The comparison between SAFRAN and $AWS_m$ incoming long wave radiation indicates an overestimation of SAFRAN data for **low** cloudiness conditions. This can be explained by local orographic features and/or **high**-altitude clouds that are not considered in SAFRAN reanalysis." This has been changed in the new manuscript.

Given that this sentence was neither clear nor correct, we decided to rewrite it as follows: "The comparison between SAFRAN and $AWS_m$ incoming long wave radiation indicates an overestimation of SAFRAN data for **low** cloudiness conditions. This can be caused by **high**-altitude clouds that are not considered in SAFRAN reanalysis and an incorrect vertical discretization of the atmosphere in SAFRAN."

Finally, we also add the RMSE values (in accordance with comment 18) on the bias between measured and calculated incoming LW as follows: "Using this correction, the correlation between $AWS_m$ incoming LW radiation and corrected LW radiation from SAFRAN was increased from $R^2 = 0.71$ to $R^2 = 0.83$ and the RMSE decreased from 44.3 W m$^{-2}$ to 29.7 W m$^{-2}$."

**Specific comment #20 (P7 L17):**

*How do the estimates of incoming (slope-corrected) shortwave from SAFRAN compare with the values you calculate from the observed reflected shortwave and the albedo scheme?*

**Authors' response:**

First, we propose to add details concerning the comparison of incoming SW from SAFRAN and AWS measurements:

"Correlations between daily incoming shortwave radiation ($R^2 = 0.81$) are significant at the 99% confidence level (Student's t test) and RMSE = 77.2 W m$^{-2}$."

Note that this comparison was done using AWS measurements and the incoming SW from SAFRAN after processing in Crocus, which takes into account the topography (slope, exposure…). This incoming SW was used in particular to compute the albedo. In addition, measured and simulated albedo were compared (data not reported to avoid information overload). Nevertheless in the new version more information about albedo calibration and validation is provided:

In the model description (section 3.1.1, Crocus model), the text now reads:

"Note that albedo measurements performed at $AWS_{g06}$, $AWG_{g08}$ and $AWS_{g-accu08}$ were used to calibrate and validate ice and snow albedo in the model (see section 4.2.3.2)"

In addition, in the results part (section 4.2.3.2, Ice albedo), the following information has been added:

"Albedo measurements were used to calibrate and validate the albedo range in the model. The correlation between daily albedo measurements and daily simulations is significant (95% confidence level (Student's t test), $R^2 = 0.54$, RMSE = 0.23). In particular, the transition date from snow surface to ice surface is well represented in the ablation area (difference lower than 5 days). Nevertheless, surface albedo is highly variable in time and space and validation was carried out at only two points (one in the ablation area and one in the accumulation area) over a short time period (three months). Therefore, this variable was tested to evaluate its sensitivity."

**Specific comment #21 (P7 L19):**

*What is meant by wind speed 'generally considered' to be at 2 m?*

**Authors' response:**

You are right this is confusing. Therefore 'generally' has been removed.

**Specific comment #22 (P7 L21-22):**

*- 'This underestimation is…' This statement could use a reference to supporting studies.*

*- In addition, it would be useful to present a comparison of wind direction in addition to wind speed. Is the observed wind direction in the downslope direction during these periods of large wind speed differences and suspected katabatic flow? How well does SAFRAN represent the local wind direction i.e. is the influence of the glacier accounted for?*

**Authors' response:**

- A reference to the work made by Dumont *et al.,* (2012) has been added in reponse to your comment.

- Concerning the comment on wind direction, Litt *et al.* (2017) has investigated this point and we have added the following sentence: "As mentioned in Litt *et al.* (2017), when large-scale atmospheric forcing was strong, intense downslope winds were observed, aligned with the main glacier flow (*i.e.* coming from the South, see Figure 1)."

**Specific comment #23 (P9 L4-5):**

*Comma required between 'model, implemented' and '2013), was'.*

**Authors' response:**

Done

**Specific comment #25 (P9 L9-10):**

*How are the empirical values selected for microstructure? Are they glacier specific? What is the uncertainty in them?*

**Authors' response:**

A sentence as been added as recommended:

"The variables are grain size/dendricity and sphericity (see Vionnet et al., 2012, for more details). Their values are specified for glacier ice. The specified values only impact the calculation of albedo/light penetration depth, which is constant for ice."

**Specific comment #26 (P9 L17):**

*The density value of 917 kg m-3 is generally assigned to pure glacier ice. Have you examined your model's sensitivity to varying this value i.e. to account for uncertainty in the actual ice density?*

**Authors' response:**

This is a good point. Such sensitivity tests were performed by our colleagues using Crocus on a glacier in New Zeeland. Different tests were performed using a pure glacier ice density (917 kg m$^{-3}$), a 900 kg m$^{-3}$ value and a smaller density (850 kg m$^{-3}$). They found that the melt sensitivity to density is very low compared to the melt sensitivity to albedo or surface roughness (difference of less than 5% of the melt at the end of summer season between 917 and 850 kg m$^{-3}$. Unpublished results). Based on this work, we considered not changing the density value used in the model.

**Specific comment #27 (P10 L10):**

*When running the ATI, what data are used for the incoming potential direct solar radiation (IPOT)? Is this simply the incoming direct shortwave radiation from SAFRAN?*

**Authors' response:**

IPOT is the potential, clear-sky direct solar radiation calculated following Hock (1999). This

information and the reference have been added. You can now read: "… and *IPOT* is the potential, clear-sky direct solar radiation (W m$^{-2}$) calculated following Hock (1999)."

**Specific comment #28 (P10 L16):**

*I know the ATI model uses the WSMB from crocus, but is summertime precipitation and potential summertime snowfall accounted for when using the ATI?*

**Authors' response:**

This has been clarified as follows in response to your comment: "Note that in the ATI model, summertime snowfalls are deduced from SAFRAN data."

**Specific comment #29 (P10 L31- 32):**

*'Peformance was evaluated…' Strictly speaking, you are not evaluating ATI performance with winter SMB measurements as you are using Crocus WSMB values (as mention above in L26 -27). Consider restructuring this sentence.*

**Authors' response:**

The sentence has been re-written in response to your comment: "Performance was evaluated by comparing summer SMB simulated by the ATI and Crocus models to summer SMB measurements of each stake located in the ablation and accumulation areas."

**Specific comment #30 (P11 L33):**

*Typo; missing an 'in' between presented and section, and section number should be 4.2.3.*

**Authors' response:**

Done

**Specific comment #31 (P12 L22- 24):**

*'Indeed, the ATI…' This sentence needs to be rewritten, and the figure reference should be Figure 4a. Looking at Figure 4a, when observed SSMB is above -2 m w.e. (i.e. less negative than -2), the ATI model shows a tendency to underestimate SSMB (more negative), or in other words, to overestimate ablation.*

**Authors' response:**

The sentence has been rewritten in reponse to your comment. You can now read:

"Indeed, the ATI model under-estimated the SSMB values, when observed SSMB is above -2 m w.e., and in particular those corresponding to the accumulation area (Figure 4a)). This leads to a significant decrease in the correlations between measurements and simulations."

**Specific comment #32 (P12 L26-28):**

*'In addition, the temporal evolution…'*

*- By temporal evolution, do you mean you compared the cumulative simulated surface mass balances from the two models over the summer?*

*- If so, was this using a daily time step?*

*- Was this carried out just for one summer season or all?*

*- You have presented the maximum differences for the SSMBs, but perhaps if you are interested in the temporal evolution, it would be more interesting to describe the biases/differences that occurred within the season (e.g. are the SSMB differences driven by a general bias or by individual days with large differences etc).*

**Authors' response:**

- Yes, this is exactly what has been done: we compared the cumulative simulated mass balance over the hydrological year (as was done in Figure 6 for the sensitivity tests)

- Yes, we did that using a daily time step.

- It was done for 10 years (2006-2015) to see if there were systematic differences between models.

In response to your comment and to provide more details concerning the results, this information as well as a new figure (Figure 5) with a better description have been added to the manuscript. You can now read:

"In addition, the temporal evolution of the simulated summer SMBs over season is shown in Figure 5. Daily summer SMB data simulated by both models are reported in each graph for different measurement points. Note that this was done for all years over the period 2006-2015, but only results for the year 2008 are represented here for the sake of clarity. For this year, results indicate very similar performance for the two models in the lower part of the ablation area (stakes 1 to 22) at the end of the season: the absolute mean difference of summer SMB is 0.13 m w.e. yr$^{-1}$ (lower than the measurement uncertainty) and the maximum difference is 0.36 m w.e. yr$^{-1}$. The same is true when we consider all the ablation season results (i.e. 2006-2015): the maximum difference of 0.45 m w.e. yr$^{-1}$. For the year 2008, the ATI model simulates lower ablation compared to Crocus model during August in the lowest part of the glacier (e.g. stakes 20 and 22), and higher ablation during October (Figure 5). However these results are specific to this year and these stakes. No systematic difference is observed.

In the accumulation area and close to the equilibrium line (e.g. stakes 27 or 30), differences between the summer SMBs simulated by the two models (Figure 5) are greater: the absolute mean difference of summer SMB is 0.56 m w.e. yr$^{-1}$ and the maximum difference is 0.87 m w.e. yr$^{-1}$. Considering all the years, the maximum difference is 1.91 m w.e. yr$^{-1}$. Here again, there is no systematic difference, except that maximum differences are generally observed in June and October."

[Figure]

**2008**

Figure 5: Surface mass balance evolution at some selected measurement points of Saint-Sorlin glacier over the hydrological year 2007-2008. Black lines represent the simulated SMB using the Crocus model with corrected forcing. The blue curves show the simulations made using the ATI model with simulated winter SMB adjusted by measurements. Black dots represent the measurements with their uncertainties.

**Specific comment #33 (P12 L30):**

*'Here again…' Following on from my point above for L22 – 24, SSMBs in the accumulation area from the ATI model appear to be underestimated; it is the ablation that is overestimated.*

**Authors' response:**

This has been corrected in response to your comment. You can now read:

Here again, summer SMBs simulated with the ATI model in the accumulation area are under-estimated.

**Specific comment #34 (P13 Figure 3):**

*Axis labels should specify surface mass balance (SMB) not MB.*

**Authors' response:**

Done

**Specific comment #35 (P14 Figure 4):**

*Same point as for fig 3.*

*- Also, in caption, using 'a-c' etc. suggests a to c, inclusive (i.e. a,b,c). Consider using 'a and c'.*

**Authors' response:**

- Done

- In caption 'and' has been used instead of '-'in reponse to your comment.

**Specific comment #36 (P14 L8-10):**

*'various stakes'. How many stakes was this performed on? Some results/plots from other stakes would be useful. Maybe provide the mean and standard deviation (over all tested stakes) of the differences in annual SMBs for the extreme summers and winters mentioned here.*

**Authors' response:**

We agree with this comment and have added more information in response to your comment. The sentence has been reworded:

"The tests (described in section 3.2.1) of the annual mass balance sensitivity to seasonal mass balance using the Crocus model were performed at seven stakes in the ablation area, ranging between 2700 m a.s.l. and 2870 m a.s.l. For the sake of clarity, only the results for stake #10 (located at 2760 m a.s.l.) are presented in Figure 6, but conclusions are similar for all the stakes."

Additional results have been added later in this section in response to your comment:

"The difference in annual SMBs between these two extreme summers for the stake #10 was 4.1 m w.e. yr$^{-1}$ at the end of the hydrological year. Similar results are found for the other stakes: the mean difference is 4.4 m w.e. yr$^{-1}$ with a standard deviation of 0.41 m w.e. yr$^{-1}$.

The sensitivity of annual SMB to winter SMB is illustrated by Figure 6b. Note that for the sake of clarity, only the two extreme years of the time series (2000-2001, highest winter SMB (pink line) and 2008-2009, lowest winter SMB (blue line)) are presented in Figure 6b. The difference between these two years on 15 April is 1.2 m w.e at stake #10 (and on average 1.1 m w.e with a standard deviation of 0.13 m w.e considering all the stakes). Using the same summer conditions, the difference at the end of the hydrological year is 2.4 m w.e. (*i.e.* twice the difference at the end of the winter season). Here again results are similar for the all the stakes considered: the mean difference is 2.2 m w.e. yr$^{-1}$ with a standard deviation of 0.21 m w.e. yr$^{-1}$.

The same test was performed using the extreme 2003 summer conditions instead of the mean summer conditions. In this case, the difference at the end of the hydrological year for all the stakes was considerably larger (3.4 m w.e., in mean, standard deviation 0.45 m w.e. yr$^{-1}$; results not shown). These results confirm that the annual SMB variability is mainly driven by the summer SMB variability (*i.e.* differences are larger when we considered a mean winter and all the summer conditions than the

contrary). Nevertheless, the annual SMB appears to be very sensitive to the winter SMB, in particular for extreme years."

**Specific comment #37 (P15 L5):**

*Have you quantified this as 'significant'?*

**Authors' response:**

As is difficult to evaluate if it is statistically significant in this case, the sentence has been reworded:

"Nevertheless, the annual SMB appears to be very sensitive to the winter SMB, in particular for extreme years."

**Specific comment #38 (P15 L23):**

*'First, correlations were computed…' Correlations with what?*

**Authors' response:**

The following explanation has been added:

"First, correlations between simulated and measured summer SMB were computed for all the stake measurements (in the accumulation and ablation areas)…"

**Specific comment #39 (P17 L1-4):**

*Repeated referencing of table 3 is probably unnecessary in the same subsection.*

**Authors' response:**

Done. Only one reference has been kept.

**Specific comment #40 (P17 Table 4):**

*Caption appears to be the same as that used for table 3.*

*- Why are the performances for ASMB without LW and wind corrections so poor compared to the seasonal performances?*

**Authors' response:**

- Caption has been corrected as follows in Table 4:

"NS efficiency coefficients for simulated surface mass balances with respect to measured values over the 2006-2010 period. Simulations were performed with and without correction of the meteorological forcing from SAFRAN."

- This was a mistake that slipped into the original manuscript. All the Nash-Sutcliff coefficients mentioned in this table have been carefully re-computed and you can now read:

| | $NS$ for ASMB | $NS$ for WSMB | $NS$ for SSMB |
|---|---|---|---|
| SAFRAN with corrected data | 0.67 | 0.78 | 0.86 |
| Without LW correction | 0.36 | 0.73 | 0.65 |
| Without wind speed correction | 0.27 | 0.59 | 0.71 |

**Specific comment #41 (P18 Figure 6):**

*It would be interesting in these plots to see the effect of removing the precipitation correction also.*

**Authors' response:**

Considering that the goal of the paper is to test the performance of different models to simulate the snow/ice melt and because SAFRAN is well known to under-estimate precipitation at high altitudes, we have considered that showing the effect of removing the correction precipitation will not add valuable information in the paper. Correction are needed and according to us, the 4.2.2.1 section (sensitivity to precipitation correction) describes in detail which correction is the best to correct precipitation over the glacier.

**Specific comment #42 (P18 L11):**

*Swap the position of 'blue' and 'black' to avoid confusion.*

**Authors' response:**

Done

**Specific comment #43 (P18 L13):**

*The use of…' Ensure that you refer to correcting the wind speed data rather than 'wind data', as you have not discussed correcting the wind direction in this paper. The 'Without wind correction' label in Table 4 should also be corrected.*

**Authors' response:**

'wind data' was replaced by 'wind speed data' in response to your comment.

Table 4 has also been corrected.

|  | $NS$ for ASMB | $NS$ for WSMB | $NS$ for SSMB |
|---|---|---|---|
| SAFRAN with corrected data | 0.67 | 0.78 | 0.86 |
| Without LW correction | 0.36 | 0.73 | 0.65 |
| Without wind speed correction | 0.27 | 0.59 | 0.71 |

**Specific comment #44 (P19 L5):**

*Provide mean values of the turbulent fluxes using observed and SAFRAN wind speeds.*

**Authors' response:**

These information have been added according to your comment:

"Actually each component of the turbulent fluxes (H and LE) simulated with original SAFRAN wind data is lower than those simulated with the measured wind. For instance, the mean value of H computed over summer 2006 is equal to 7.2 W m$^{-2}$ (with a standard deviation of 10.7 W m$^{-2}$) when simulated with SAFRAN wind data, compared to 22.2 W m$^{-2}$ (with a standard deviation of 37.8 W m$^{-2}$) when using measured wind speed.

**Specific comment #45 (P19 L11-12):**

*Replace 'up to' (e.g. 'up to 20 C') with 'max. increase of'. The use of 'up to' here sounds like the surface temperature is increased to this temperature!*

**Authors' response:**

Done

**Specific comment #46 (P19 L21):**

*Can you clarify what you mean by a positive feedback in this case? An increase in ice/snow surface temperature can reduce the turbulent heat flux into the surface.*

**Authors' response:**

The sentence "Note that the surface temperature also has a positive feedback on the turbulent fluxes, leading to a complex relation between these variables." has been modified and now reads:

"Note that the surface temperature also has a feedback on the turbulent fluxes (e.g. an increase in ice/snow surface temperature can reduce the turbulent heat flux into the surface), leading to a complex relation between these variables."

**Specific comment #47 (P19 L23-24):**

*- Add speed to '…correction of wind on the winter…'*

*- Also, just to clarify, does wind speed not affect how the model deals with snow accumulation?*

**Authors' response:**

-Done

- We are not sure that we understand your question. If the question is "Is wind speed taken into account in the distribution of the solid precipitation in SAFRAN", the answer is no. However, because snow accumulation is specified by measurements in our simulations, the spatial variability of the winter balance due to wind transport is considered. Therefore, there is no impact on simulated summer SMB.

**Specific comment #48 (P19 L26-27):**

*'A larger impact…' Are you suggesting this is due to it being an ice surface or because the surface is warmer at this time and melt is occurring?*

**Authors' response:**

We observed larger differences when the snow is gone and the surface is ice. The sentence has been change and now reads:

"A larger impact of the correction of wind on SMB can be observed during the second part of the summer in the ablation area (from about July, Figure 7) when the surface is ice, indicating the importance of having wind speed measurements to compute turbulent fluxes."

**Specific comment #49 (P20 Figure 7):**

*Explain dashed line (i.e. melting point).*

**Authors' response:**

We have added: "Dashed line corresponds to the melting point." in the caption of the figure (now figure 8).

**Specific comment #50 (P20 L8-9):**

*Why only test for roughness values that were larger than the employed roughness value?*

**Authors' response:**

Initial roughness values in Crocus are $Z0ice = 1$ mm and $Z0snow = 0.5$ mm. Considering that the values generally found in the literature for ice surfaces are most of the time in the range of 1 - 6 mm (Brock *et al.,* 2006; Smith *et al.,* 2016) and can reach 80 mm for very rough glacier ice (Smeets *et al.,* 1999), we thought that 1, 10 and 100 mm for ice roughness lengths were useful tests. We however agree with your comment, that lower values should be tested in the future. Considering that roughness increases during summer season and because the objective of this figure was to show the impact of a changing roughness length, no further tests were performed. Nevertheless this question should be investigated in further studies.

In response to your comment, more details concerning the choice of these values have been provided in the manuscript as follow:

"While feedback loops exist between turbulent fluxes and surface temperature, we attempted to assess the impact of effective roughness length values (varying arbitrarily from a factor 1 to 100) on both surface temperature and summer SMB. Considering that the values generally found in the literature for ice surfaces are most of the time in the range of 1 to 6 mm (e.g. Brock *et al.,* 2006, Table 1; Smith *et al.,* 2016) and can reach 80 mm for very rough glacier ice (Smeets *et al.,* 1999), the tests were performed with ice values ranging between 1 and 100 mm (Figure 9)."

Note that references have been added in the manuscript:

Brock, B.W., Willis, I.C. and Sharp J.M.: Measurement and parameterization of aerodynamic roughness length variations at Haut Glacier d'Arolla, Switzerland. J. Glaciol. 52, 281–297. doi:10.3189/172756506781828746, 2006

Smeets, C. J. P. P., Duynkerke, P., and Vugts, H.: Observed wind profiles and turbulence fluxes over an ice surface with changing surface roughness. Boundary-Layer Meteorology, 92(1), 99–121, 1999

Smith, M.W., Quincey, D.J., Dixon, T., Bingham, R.G., Carrick, J.L., Irvine-Fynn, T.D.L. and Rippin, D.M. : Aerodynamic roughness of ice surfaces derived from high resolution topographic data. J. Geophys. Res., 121(4). DOI: 10.1002/2015JF003759, 2016

**Specific comment #51 (P20 L10-13):**

*-It would be useful to present the values for the % difference in SMB between the different roughness value scenarios, as it is hard to distinguish in figure 8 if there really are different responses for snow and ice surfaces, or if it just that the magnitudes are greater because there is greater levels of ablation over the ice surfaces.*

*-Clarify what you think your results are showing. Are you suggesting that varying the implemented roughness lengths for snow will not affect the turbulent heat fluxes estimated by the model over this surface? This would not follow the theory of the bulk aerodynamic method implemented in the model.*

**Authors' response:**

- First, the impact of changing roughness length has been clarified in the text, as follow:

"Figure 9a illustrates a stronger impact for more negative SSBMs (corresponding to mainly ice ablation) than for the less negative SSMBs (corresponding to snow ablation). This is confirmed by results shown in Figures 9b (one stake in the ablation zone) and 9c (one stake in the accumulation area). At the end of the hydrological season, there is a difference of SMB of 77% in the ablation area (ice surface) and 11% in the accumulation area (snow surface) between roughness values of 1 and 100."

- Later in the paragraph, we clarified our results. Changing roughness length clearly affects turbulent fluxes, both over snow or ice surfaces. We however observed that this effect is more pronounced over icy surfaces. The impact of changing roughness length is limited on snow ablation considering that albedo changes have greater impacts. The sentence was reworded:

"In fact, changing the roughness length considerably affects the simulated ice ablation (Figure 9b) but the effect is limited on the simulated snow ablation (Figure 9c), considering that the impact of other parameters such as albedo changes can be greater."

**Specific comment #52 (P20 L17-18):**

*'Note that…' Was this a single sum over the full season, or did you examine the simulated fluxes over shorter timescales e.g. daily sums? Using shorter timescales might allow you to look for the temporal variation in roughness length you have mentioned.*

**Authors' response:**

The sentence is : « Note that $z_0$ was calibrated by fitting the simulated sum of H and LE with the one calculated with the EC method."

Yes, $z_0$ was calibrated using observations over the full season, as Eddy covariance observations are available only over a short summer. The publication by Litt *et al.* (2017) investigated the different methods to determine $z_0$ (wind mast, Eddy covariance, bulk method) and associated uncertainties. They also analyzed daily evolution of turbulent fluxes, in relation to wind speed and direction.

A detailed study of the impact of surface roughness on shorter timescales is beyond the scope of this paper as our main objective is to study the relative performance of melt models over summer season. However, considering comments of reviewer 1 and the uncertainty linked to surface roughness length on turbulent fluxes, we reworded the sentence later in the manuscript as follow :

"As $z_0$ can vary considerably over time (including at daily timescales) and space (snow or ice surfaces), and due to the strong sensitivity of the model to this parameter and the large uncertainty in its determination, having *in-situ* turbulent flux measurements over ice and snow surfaces, and covering various summer ablation seasons is very useful to properly calibrate $z_0$."

**Specific comment #53 (P20 L20):**

*You have arbitrarily selected the values for the scalar roughness lengths to be 1/10th of that for momentum. However, there are other schemes suggested in the literature to estimate the scalar values, such as assuming a 1/100 ratio, assuming they are all equal, and utilising a surface renewal method (Andreas, 1987). Have you considered the model sensitivity to these values? Another major source of uncertainty in the estimating the turbulent fluxes is how the model accounts for changes in atmospheric stability. While an investigation of this uncertainty may be beyond the scope of this paper, it is worth mentioning it as a source of uncertainty, particularly when using reanalysis data where the localised effects of the glacier on stability may not be resolved. Andreas, E. L. (1987). A theory for the scalar roughness and the scalar transfer coefficients over snow and sea ice. Boundary-Layer Meteorol. 38, 159–184. doi: 10.1007/BF00121562*

**Authors' response:**

We agree with the reviewer, one can question the fact that the parameterization of the scalar roughnesses has been chosen as an arbitrary function of the momentum roughnesses. Thought, this parameterization consists in a linear function. Since the turbulent fluxes in the model are calibrated to correspond to the EC measurements, a change in this linear function (or an equality) would anyway lead to a different calibration of the momentum roughnesses in order for the whole turbulent fluxes to correspond to the measurements. That is, it corresponds to calibrating only one effective roughness, so that would have limited impact anyway. Nevertheless, the reviewer's comment is relevant: for periods outside the calibration period the choice of a single effective roughness can impact the fluxes evaluation, but since this choice works well to reproduce the fluxes during the calibration this assumption still makes sense. Furthermore the sensitivity of the fluxes and the modelled mass balance to the choice of the roughnesses scheme has been tested in the new submitted version and we underline its high importance (section 4.2.3.1). Testing the surface renewal method would clearly improve the roughnesses estimation, but we think this goes beyond the scope of the study. We nevertheless now acknowledge it in the manuscript (Andreas, 1987). Similarly, stability corrections would certainly be necessary and a clear improvement for such study (probably reducing the magnitude of the fluxes). As suggested by the reviewer we mention this issue in the manuscript as follows:

"Uncertainty evaluation should be further investigated, for instance, by testing the surface renewal method to improve the scalarroughnesses estimation, or in considering stability corrections (Andreas, 1987)."

**Specific comment #54 (P21 Figure 8):**

*Reposition '(b)' and '(c)' in the caption to be in front of the relevant text.*

**Authors' response:**

Done

**Specific comment #55 (P21L14):**

*This finding regarding the importance of ice albedo to model performance may not be very transferable. The same test performed on a glacier in a different climate setting (e.g. one where radiation has a larger contribution to melt energy) may find model performance to be more sensitive to albedo parameterisation.*

**Authors' response:**

We agree with this comment, and also considering comment 56, we have added the following information:

"Our results point out that this parameter needs to be properly calibrated with measurements on Saint-Sorlin glacier to optimize model performance. However, model performance is possibly more sensitive to albedo parameterization for glaciers where radiation has a larger contribution to melt energy. Six *et al.* (2009, Table 3) showed that, during summer 2006, monthly mean fluxes of the

energy balance were 80% for the net all-wave radiation R and 20% for the turbulent fluxes. This monthly mean distribution can reach 70% and 30% respectively on a daily time scale. For Zongo Glacier (Bolivia, 16° S) the net all wave radiation R can represent 97% of the energy balance (Sicart *et al.,* 2008)."

**Specific comment #56 (P22 L31, P23 L2):**

*Following on from the point above, it would be interesting to see the partitioning of the melt energy for the study glacier, and a summary of the meteorological conditions. These model sensitivity findings may be very dependent on the ratio between the turbulent heat and radiation fluxes.*

**Authors' response:**

As also requested by reviewer 2, we first propose to add a summary of meteorological conditions in a supplementary material. Based on 8 years of $AWS_m$ records on the moraine (2006-2013), daily means of temperature (°C), relative humidity (%), incident shortwave radiation (W/m²), incoming longwave radiation (W/m²), wind speed (m/s) are presented on the figure (see below). Concerning wind direction, instantaneous half hourly data are classified and a percentage of number of data per direction is given.

We proposed to add in the manuscript, ligne119:

"A summary of the meteorological conditions at $AWS_m$ is given in the supplementary material"

The related figure and the figure caption are:

[Figure]

Figure 1: Daily means of a) air temperature (°C), b) relative humidity (%), c) wind speed (m/s), d) incident shortwave radiation (W/m²), e) incoming longwave radiation (W/m²), f) percentage of instantaneous half hourly wind direction data in each direction (%). Data are calculated on the 2006-2013 period, at $AWS_m$ (2720 m)

In addition, you also mentioned the possibility to see the partitioning of the melt energy. Different works have been done on Saint-Sorlin and we propose to refer to these previous studies (Six *et al.,* 2009; Sicart *et al.,* 2008; Dumont et al., 2012). In summer 2006 (Six et al., 2009, table 3), monthly mean measurements of the different fluxes of the energy balance were divided as follow: 80% for the net all-wave radiation balance and 20% for the turbulent fluxes. This monthly mean distribution can reach 70% and 30% respectively on a daily time scale. These results were added in the answer you're your comment #55.

Finally, as you mention, these model sensitivity findings may be very dependent on the ratio between the turbulent heat and radiation fluxes. In Sicart *et al.* (2008), they found that at Sorlin, the turbulent sensible heat flux is greater than in Bolivia, because of higher temperatures, but melt variability is still

controlled by short-wave radiation. This point was already addressed in comment's answer #55 and the 4.2.3.2 section has been completed as follow:

"Our results point out that this parameter needs to be properly calibrated with measurements on Saint-Sorlin glacier to optimize model performance. However, model performance is possibly more sensitive to albedo parameterization for glaciers where radiation has a larger contribution to melt energy. Six *et al.* (2009, Table 3) showed that, during summer 2006, monthly mean fluxes of the energy balance were 80% for the net all-wave radiation R and 20% for the turbulent fluxes. This monthly mean distribution can reach 70% and 30% respectively on a daily time scale. For Zongo Glacier (Bolivia, 16° S) the net all wave radiation R can represent 97% of the energy balance (Sicart *et al.,* 2008)."

**Response to the anonymous referee #2**

**Main comment:**

*The paper by Réveillet and others describe the results of the modeling of the surface mass balance of Saint-Sorlin glacier in the French Alps. The SMB is calculated with CROCUS, an energy balance model originally designed to calculate snow pack. This model is forced with SAFRAN reanalysis data that is corrected to the in-situ weather observation at the location of the glacier. The authors explore the sensitivity of the simulated SMB to different components of the climatic forcing and to various model parameters. Furthermore is the result of the energy balance model compared with the results of an empirical model. The study is well written and the results are interesting. I have the following comments, spit up in general comments, specific points and some technical points, the latter two per page and line number.*

**Authors' response:**

Thank you for your positive comment on the interest of the paper. We carefully answered point by point to your general comments and specific/technical points. Note that replies from the authors are in green.
* * *
**General comments**

**General comment #1:**

*The difference between energy balance model and empirical model is not just the model formulation, but also the way the models are calibrated. The parameters of the empirical model are calibrated by fitting calculated SMB to the observed SMB, while for the energy balance the meteorological input of the model is fitted to observed weather. Would the energy balance model perform better if its model parameters and input corrections are optimized to the observed SMB?*

**Authors' response:**

You are right, the performance of the energy balance in modelling SMB could be improved by using SMB measurements for calibration (as for example in calibrating the $z_0$ to fit the SMB to observations as it has been done in several studies). However, this doesn't mean that the model is performing better for good reasons. SMB can be well simulated by compensating bias and/or uncertainties (as can be the case for the turbulent fluxes as mentioned above). In this study, we have many energy balance measurements, including turbulent fluxes. Our objective is to evaluate the performance of the model while ensuring that all energy components are well modeled. This is indicated in the introduction: "Then, the surface energy and mass balance model is calibrated using the measured energy fluxes to ensure that all the energy balance components are accurately represented."

The comparison between the energy balance model and the empirical model is done to evaluate the performance of these models considering the available data. The main idea is to identify and discuss the most important variables for each model and to provide clues as to the best approach to be used for future simulations, while limiting uncertainties.

**General comment #2:**

*I do not fully see how the conclusion that empirical models would be better suited to model glacier SMB for periods, or glaciers, without AWS measurements.*

*The results of this paper show that the energy balance model performs better, mainly in the accumulation area, than the empirical model. This is also true for the period where the input of the*

*model is not directly corrected with the AWS data. So therefore I do not see why the empirical model should be more suited to model future glacier mass balance, when also no direct correction of the meteorological forcing with in-situ observation is possible.*

**Authors' response:**

We do not really agree with your comment; our results do not indicate that the energy balance model performs better than empirical approaches. Regarding the comparison between these approaches, results depend on the area (ablation *vs* accumulation) and the quality of the forcing. We have to note that this comparison have been done using an adjusting forcing (even over the period when no AWS are available to assess this correction).

Nevertheless, the study has been designed especially to assess the performance of energy balance and empirical models. For this purpose, we first evaluate the performance using the most accurate forcing we have (i.e. reanalysis adjusted with AWS measurements). Then, we performed a thorough study of sensitivity to meteorological inputs (section 4.2) and other parameters (surface roughness, …). Our modelling experiments and comparison with the *in-situ* measurements reveal a strong sensitivity of energy balance model to the wind speed, especially in the ablation area (i.e. over ice surfaces). Indeed, the use of uncorrected wind data (i.e. coming from SAFRAN reanalysis) leads to large differences in annual mass balance of the ablation area (1 to 1.7 m w.e. yr$^{-1}$), as shown in Figure 7 (blue curves). Note that the differences are smaller in the accumulation area (over snow surface). In addition, for the whole dataset of mass balance obtained from ablation and accumulation areas, the model performance (using the Nash and Sutcliffe coefficient) decreases from 0.67 to 0.27, with or without the corrected values of the wind speed data respectively. Similar conclusions are obtained for the long-wave radiation correction (please, refer to section 4.2.2.2). Although the sensitivity is low in the accumulation zone, these results clearly show that *in-situ* meteorological data are needed to ensure a good performance of energy balance model, as using reanalysis without adjustment considerably decrease the model performances. Consequently, the energy balance model cannot be transferred to another glacier without *in-situ* meteorological data. In addition, the current bias in reanalysis data highlights the complexity to model all the meteorological variables in the present; we thus expect to also have bias in future projections. In this case, and due to its strong sensitivity, an energy balance model could lead to significant bias for future simulations.

Otherwise, our study reveals a strong impact of roughness value on ice ablation and indicates the necessity of measurement to calibrate this parameter. Given that the surface roughness value is unknown in the future, this is a strong limitation of this kind of model.

Therefore, empirical approaches based on precipitation and temperature only, could be more appropriate for simulations of glaciers in the future, as performances presented in this paper are similar than for an energy balance model, in the ablation area. For the sake of clarity and to take into account this comment, the manuscript has been revised (see below).

However, despite the limitations of energy balance model for simulations in the future, this physical model remains crucial to study the processes to understand the physical relationships between the meteorological variables and ablation. In the conclusion, we added a sentence in this direction to mitigate the obstacles of energy balance model abilities to simulate the glacier mass balance in the future.

Changes made in the manuscript indicated below.

1: In the abstract section as follows:

[revised manuscript text omitted]

**General comment #3:**
*It would be instructive include information and figures on the climate at Saint-Sorlin Glacier in the paper. And to include in these figures a comparison between the original climate given by the SAFRAN data and the climate as measured with the AWS.*

**Authors' response:**

As also requested by reviewer 1, we propose to add a summary of meteorological conditions in a supplementary material. We choose to present the measured conditions on the moraine at 2720 m. Comparison with SAFRAN data are given all along the text, in section 2.3.4.

Based on 8 years of $AWS_m$ records on the moraine (2006-2013), daily means of temperature (°C), relative humidity (%), incident shortwave radiation (W/m²), incoming longwave radiation (W/m²), wind speed (m/s) are presented on the figure (see below). Concerning wind direction, instantaneous half hourly data are classified and a percentage of number of data per direction is given.

We proposed to add in the manuscript, l.119 :

"A summary of the meteorological conditions at $AWS_m$ is given in the supplementary material."

The related figure and the figure caption are:

[Figure]

**Figure 1:** Daily means of a) temperature (°C), b) relative humidity (%), c) wind speed (m/s), d) incident shortwave radiation (W/m²), e) incoming longwave radiation (W/m²), f) percentage of instantaneous half hourly wind direction data in each direction (%). Data are calculated on the 2006-2013 period, at AWS$_m$ (2720 m)
* * *
**Specific comments**

**Specific comment #1 (page 2 - l7):**
*In addition to the fact that it is questionable if the calibrated parameters are valid over long time periods, a disadvantage of the temperature-index models is that the parameters have no validity for other glaciers. Maybe you could add this here.*

**Authors' response:**
A sentence and references have been added in the introduction in response to your comment:
In addition, transferring parameters determined for an instrumented glacier to another site decreases model performance (Carenzo *et al.,* 2009; Réveillet *et al.,* 2017).

**Specific comment #2 (page 3 - l2-5):**

*Here you discuss variability, but you are, also, interested in the absolute value of the SMB, not only of the variations from year to year.*

**Authors' response:**

Right. This is the reason why the beginning of the sentence is "the temporal variability of the annual SMB is mainly driven by summer SMB variability ». We also have precised that the simulation of summer SMB (absolute value) strongly depend on winter SMB.

According to your comment, we reworded as follow:

"In the Alps, the temporal variability of the annual SMB is mainly driven by summer SMB variability (e.g., Six and Vincent, 2014). For this reason, many studies have focused on ablation modelling. However, simulated summer SMB and associated uncertainties strongly depend on the winter SMB (Réveillet *et al.*, 2017), highlighting the need for a quantification of the sensitivity of annual SMB to both seasonal components."

**Specific comment #3 (page 3 - l20):**

*I do not see that the aspects of Saint-Sorlin vary a lot in Figure 1, it is mainly N - NE - E slopes. Please be a bit more specific.*

**Authors' response:**

In response to this comment as well as comment 4, a map has been added in Figure 1, that shows glacier topography with a higher resolution and the surrounding topography. In this map, the main glacier flowing lines are indicated with arrows to better represent the different glacier aspects. We chose to add a new map instead of adding information to the one on the article for sake of clarity.

[Figure]

**Figure 1.** (a) Location of Saint-Sorlin Glacier in the French Alps. French glaciers are shown in blue except for Saint-Sorlin Glacier, used for the present study, which is in red. Black lines represent SAFRAN massif outlines (adapted from Rabatel *et al.,* 2017) (b) Aerial photo of Saint-Sorlin glacier. Blue arrows indicate the three main glacier flow lines. (c) Map of Saint-Sorlin Glacier with the network of in-situ SMB measurements (blue triangles in the accumulation area and red triangles in the ablation area). Locations of automatic weather stations used in this study are represented by green circles.

**Specific comment #4 (page 4 - Figure 1):**

*Could you add to surrounding topography in Figure 1b? That could give a better understanding for*

*your discussion of wind speed and snow distribution.*

**Authors' response:**

As mentioned in the previous comment, a map representing the surrounding topography has been added.

**Specific comment #5 (page 4 - l10):**

*- What is the accuracy of the DEMs?*

*- Do they give the same values for stable ground?*

**Authors' response:**

- These DEMs were derived from aerial photogrammetry using a 10-m spatial resolution. In particular DEM for 2003 have been done based on the same method than in Thibert et al., 2008.

Thibert E., R. Blanc, C. Vincent, N. Eckert. Glaciological and volumetric mass balance measurements: error analysis over 51 years for Glacier de Sarennes, French Alps. (2008). Journal of Glaciology, 54 (186), 522-532).

In this study, performed with aerial photographs coming from the same aerial campaign, errors due to internal stereoscopic measurements and roughness have been assessed to 1.26 m and 0.35 m respectively (i.e. 1.31 once combined) and are assumed to be similar for Saint Sorlin glacier. The orientation error is 0.22 m.

- Recent studies (unpublished) compared DEM 2003 and 2014 of Saint Sorlin glacier and indicate a mean difference of 0.52 m outside the glacier. (R. Basantes, personal communication)

In our study, DEMs have been resampled to a 200m resolution DEMs. Due to the low uncertainties mentioned above, the uncertainties due to the DEM acquisitions are negligible, in particular in this case (i.e. considering the resample).

**Specific comment #6 (page 5 - l23):**

*So SAFRAN is not gridded, but has one output per mountain region as given in Figure 1a? And this is then distributed following elevation and aspect?*

**Authors' response:**

Yes, SAFRAN data are not gridded but are given for the different massifs, altitudinal ranges (every 300m) and aspects (7 orientations). This has been clarified by the following additions to the sentence:

'… that are assumed to be homogeneous within a given massif (in particular within the Grandes Rousses massif where the Saint-Sorlin Glacier is located, Figure 1a) and depend only on altitude (one data every 300 m) and aspect (7 orientations available: N, NE, NW, S, SW, SE and 'Flat').'

**Specific comment #7 (page 7 - l5):**

*High correlation doe not say it all, as you can have a significant bias while having a high correlation. Could you indicate how well the values correspond, for example with a RMSE?*

**Authors' response:**

Yes, we agree with your comment. This comment was also made by the second reviewer. Therefore, RMSE values have been computed and added in this section as follows:

- "SAFRAN and $AWS_m$ hourly air temperatures over the ablation and accumulation seasons are well correlated ($R^2$ = 0.98 (summer) and 0.99 (winter), both significant at the 99% confidence level (Student's t test), RMSE = 0.7°C (summer) and 0.76°C (winter)). Hourly SAFRAN relative humidity is also in good agreement with the $AWS_m$ data ($R^2$ = 0.74, significant at the 95% confidence level and RMSE = 13.6%.)"

- "Using this correction, the correlation between $AWS_m$ incoming LW radiation and corrected LW radiation from SAFRAN increased the correlation from $R^2$ = 0.71 to $R^2$ = 0.83 and decreased the RMSE from 44.3 W m$^{-2}$ to 29.7 W m$^{-2}$."

Note that in the previous version, the correlation regarding the LW was indicated as R and not $R^2$. This mistake has been corrected in the current version.

- "Correlations between daily incoming shortwave radiation ($R^2 = 0.81$) are significant at the 99% confidence level (Student's t test) and RMSE = 77.2 W m$^{-2}$."

- "A poor correlation ($R^2 = 0.19$, RMSE = 3.8 m s$^{-1}$) between SAFRAN wind speed (considered at 2-m)…"

- "Since the correlation between the measured wind speed on the foreland and on the glacier is high ($R^2=0.97$, RMSE=1.7 m s$^{-1}$)"

**Specific comment #8 (page 7 - l9-10):**

*How does a lack of low-altitude clouds in SAFRAN explain an overestimate of incoming long-wave radiation in SAFRAN? I would expect that low altitude clouds are warmer than high-altitude clouds and thus emit more long wave radiation, such that not including these low-altitude clouds would lead to an underestimate of the incoming long wave radiation.*

**Authors' response:**

We completely agree. This was a mistake that slipped into the original manuscript. The sentence initially read:

"The comparison between SAFRAN and AWS$_m$ incoming long wave radiation indicates an overestimation of SAFRAN data for **low** cloudiness conditions. This can be explained by local orographic features and/or **low**-altitude clouds that are not considered in SAFRAN reanalysis."

It should be, in reality: "The comparison between SAFRAN and AWS$_m$ incoming long wave radiation indicates an overestimation of SAFRAN data for **low** cloudiness conditions. This can be explained by local orographic features and/or **high**-altitude clouds that are not considered in SAFRAN reanalysis." This has been changed in the new manuscript.

Given that this sentence was neither clear nor correct, we decided to rewrite it as follows: "The comparison between SAFRAN and AWS$_m$ incoming long wave radiation indicates an overestimation of SAFRAN data for **low** cloudiness conditions. This can be caused by **high**-altitude clouds that are not considered in SAFRAN reanalysis and an incorrect vertical discretization of the atmosphere in SAFRAN."

Finally, we also add the RMSE values (in accordance with comment 7) on the bias between measured and calculated incoming LW as follows:

"Using this correction, the correlation between AWS$_m$ incoming LW radiation and corrected LW radiation from SAFRAN increased the correlation from $R^2 = 0.71$ to $R^2 = 0.83$ and the RMSE decreased from 44.3 W m$^{-2}$ to 29.7 W m$^{-2}$."

**Specific comment #9 (page 7 -l17):**

*Also here, could you give an estimate of the bias/rmse in addition to the correlation?*

**Authors' response:**

Done, please refer to the response to comment #7.

**Specific comment #10 (page 7 - l22):**

*How likely is it that you measured the katabatic wind on the site of AWSm? It is located off the glacier and well above the nearest glacier surface (glacier extends to below 2700 m and the AWSm is at 2720 m), while the katabatic wind on a small glacier can be quite shallow. If you look at the wind direction, do you then see a consistent down-slope wind?*

**Authors' response:**

This was based on other studies. This is now indicated in the paper (see below) and a sentence has been added concerning wind direction in response to your comment and those of the other reviewer.

"This underestimation is likely due to both non-consideration of katabatic wind and local effects due to orography (Dumont *et al.,* 2012). As mentioned in Litt *et al.* (2017), when large-scale atmospheric forcing was strong, intense downslope winds were observed, aligned with the main glacier flow (*i.e.* coming from the South, see Figure 1)."

**Specific comment #11 (page 10 - l30):**
*It is not clear why you perform the calculations on a 200 m resolution. Here you refer to 2.2.2, where in turn is referred to 2.3.3 where it is stated 'we linearly interpolated on the 200-m horizontal resolution grid' without any further reason why this interpolated to this 200 m. Please explain.*
**Authors' response:**
"See section 2.2.2" has been replaced by "see sections 2.2.2 and 2.3.3.
In addition, the following information regarding the reason for this grid size has been added:
"Note that a 200 m resolution was chosen as a compromise to be sufficiently precise to consider the spatial variation of Saint Sorlin glacier (in particular the variation of aspect) and capture variability between stakes, while maintaining relevance regarding the meteorological forcing (given that values are available every 300 m of elevation)."

**Specific comment #12 (page 11 - l10-33):**
*I would add the content of these section 3.2.2.x to the respective subsections in the results section 4.*
**Authors' response:**
As according us no results are presented in this section, we would like to keep this content in the method section.

**Specific comment #13 (page 12 - l10-11):**
*- Here you write that the interpolation used to create the precipitation can explain the difference between measured and simulated winter SMB. But if all winter SMB measurements are included in the determination of the precipitation fraction maps, then how does the interpolation between these points affect the model results on the stake locations?*
*- And how do melting events in the accumulation period explain the difference? They should be in the simulations as well?*
**Authors' response:**
- The interpolation method is the krigging: it allows interpolating the data with the best fit at each measurements point; but as the grid resolution is 200 m, it could exist differences between measured and interpolated points. In addition the exact position of each stake can be distinct from one year to an other (few meters), and can lead to differences.
- Regarding the melting events, you are right, it is considered in the simulations. Nevertheless, it doesn't mean that the model is able to simulate the exact rate of melting, and it therefore exist an uncertainty.

**Specific comment #14 (page 12  - ln29):**
*This contradicts the conclusion that when no correction in the forcing is possible due to lack of AWS data, the ATI performs better than the energy balance simulations*
**Authors' response:**
In fact, these results indicate that "Over the period 1996-2005, considering all the point data over the entire glacier, Crocus performs better than the ATI model". But as mention later, this is explained by the low performance of the ATI in simulating ablation in the accumulation part: "Here again, summer SMBs simulated with the ATI model in the accumulation area are under-estimated." And if we consider only the ablation part, the ATI performs better: "On the other hand, when considering the

ablation area only, results from the ATI model better fit the summer SMB measurements (NS is 0.36 for Crocus and 0.59 for the ATI model)."

Indeed, it is difficult to draw conclusions for the entire glacier, but note that these conclusions are drawn based on a correcting forcing (even if this correction hasn't been validated over this period). As mentioned in the main comment #2, the sensitivity of meteorological forcing and surface roughness is significant, and in particular for ice surface (which mean in the ablation area; see sections 4.3.2.2, 4.3.2.3 and 4.2.3.1). As these data are really difficult to be modeled in the present, it should lead to significant uncertainties in the future. Due to these uncertainties and the sensitivity of the model to these variables, we suggest that an empirical model, requiring only the temperature appears to be a good option, especially to model the summer SMB in the ablation area.

Nevertheless, in response to you comment, it has been specified in the conclusion that ATI model is probably more appropriate for futures summer SMB simulations, but in particular for the ablation area:

"Otherwise, although empirical approaches based on simple meteorological variables also have serious drawbacks, they could be more appropriate for simulations of glaciers in the future, especially to simulate summer SMB in ablation areas, bearing in mind the lack of availability of reliable information on future meteorological variables and surface roughness."

In addition some changes have been made to clarify our conclusions (see main comment #2).

**Specific comment #15 (page 14 - Figure 4):**
*You do not plot the correlation here.*
**Authors' response:**
"Correlation" has been replaced by "comparison" in response to your comment.

**Specific comment #16 (section 4.1.3):**
*I think the results of this section could be clarified with a scatter plot where you plot annual SMB vs summer SMB with constant winter SMB and one plot with annual SMB vs winter SMB for constant summer SMB.*

**Authors' response:**
In response to your comment, and to be consistent with suggestions made by the other reviewer, additional information have been added into the text. Note that we decided not to add a new figure, as there are already a lot.

It hasn't been mentioned in the previous version, but the study has been done at several stakes. Thus mean and standard deviation are now mentioned. The paragraph has been re-written as follow:

"The tests (described in section 3.2.1) of the annual mass balance sensitivity to seasonal mass balance using the Crocus model were performed at seven stakes in the ablation area, ranging between 2700 m a.s.l. and 2870 m a.s.l.. For the sake of clarity, only the results for stake #10 (located at 2760 m a.s.l.) are presented in Figure 6, but conclusions are similar for all the stakes.

Regarding the sensitivity of annual SMB to summer SMB (Figure 6a), the results show that the simulated annual SMB was the least negative with 1995 summer conditions (green curve) and the most negative with 2003 summer conditions (red line). The difference in annual SMBs between these two extreme summers for the stake #10 was 4.1 m w.e. yr$^{-1}$ at the end of the hydrological year. Similar results are found for the other stakes: the mean difference is 4.4 m w.e. yr$^{-1}$ with a standard deviation of 0.41 m w.e. yr$^{-1}$.

The sensitivity of annual SMB to winter SMB is illustrated by Figure 6b. Note that for the sake of clarity, only the two extreme years of the time series (2000-2001, highest winter SMB (pink line) and 2008-2009, lowest winter SMB (blue line)) are presented in Figure 6b. The difference between these two years on 15 April is 1.2 m w.e. at stake #10 (and on average 1.1 m w.e. with a standard deviation

of 0.13 m w.e. considering all the stakes). Using the same summer conditions, the difference at the end of the hydrological year is 2.4 m w.e. (*i.e.* twice the difference at the end of the winter season). Here again results are similar for the all the stakes considered: the mean difference is 2.2 m w.e. yr$^{-1}$ with a standard deviation of 0.21 m w.e. yr$^{-1}$.

The same test was performed using the extreme 2003 summer conditions instead of the mean summer conditions. In this case, the difference at the end of the hydrological year for all the stakes was considerably larger (3.4 m w.e. in mean, standard deviation 0.45 m w.e. yr$^{-1}$; results not shown). These results confirm that the annual SMB variability is mainly driven by the summer SMB variability (*i.e.* differences are larger when we considered a mean winter and all the summer conditions than the contrary). Nevertheless, the annual SMB appears to be very sensitive to the winter SMB, in particular for extreme years."

In addition we think that this figure shows clearly the entire range of the sensitivity, and id easy to read. We worry about confusing thing with point representation.

**Specific comment #15 (page 16 - l6-9):**

*This part is not so clear to me. How can differences up to 25% be explained by 'only slightly affect the simulated SMB for a limited number of stakes'?*

**Authors' response:**

We agree with your comment, this point is unclear in the current version. Indeed, the maximum is 25%, which is quite high. Nevertheless, it is only affect few stakes (maximum 5) and the mean (also mentioned in this paragraph) is much lower (and generally lower than the measurement uncertainty). Our conclusions saying that it only "slightly affect the simulated SSMB" have been drawn considering this mean. This has been re-written to make it clearer:

"The highest differences between simulations and measurements are obtained for the stakes located in the lower part of the glacier tongue, using 1998 and 2007 DEMs (*i.e.* where geometric changes are the greatest). Simulations performed with 1998 and 2007 DEMs led to a mean difference in simulated summer SMBs of 0.19 m w.e. yr$^{-1}$ (~5% of the SSMBs) and reached 0.64 m w.e. yr$^{-1}$ for the lowest stakes (~15% of the summer SMBs and ~20% of the annual SMBs). Simulations performed with 2007 and 2014 DEMs, led to a mean difference of 0.15 w.e. yr$^{-1}$ (<5% of the SSMBs) and a maximum of 0.47 5 w.e. yr$^{-1}$ for the lowest stakes. Note that the differences in simulated summer SMBs *vs.* measurements in the accumulation area are larger when considering the DEMs from 2014 and 2007 than with 1998 and 2007 DEMs and can reach 0.38 m w.e. yr-1 (~20% of the summer SMBs and ~25% of the annual SMBs). Despite changes in glacier surface topography over the entire study period, such changes only affect the simulated summer SMB (*i.e.* considering changes larger that measurement uncertainty) for a limited number of individual stakes (maximum 5). Considering the entire glacier, these changes in the simulated summer SMB are negligible as the mean is lower than the measurement uncertainty."

**Specific comment #16 (page 23 - l24):**

*I do not really understand the conclusion that empirical models would be better fitted to model future SMB. From your results it is clear that using observations to correct forcing the energy balance model performs better than the empirical model, also for the period where no AWS is available. It is unclear whether the forcing corrections remain valid in the future, but the same holds for the parameters in the empirical model (as you have pointed out). So why is then the empirical model more reliable for future projections?*

**Authors' response:**

Please refer to our answer to your main comment #2 and the specific comment #14.
* * *
**Technical points**

**Technical point #1 (page 1):**
*Why not include Saint-Sorlin in the title, replacing the French Alps?*
**Authors' response:**
We agree and have changed the title:
"Relative performance of empirical and physical models in assessing the seasonal and annual glacier surface mass balance of Saint Sorlin glacier (French Alps)."

**Technical point #2 (page 3 - l23):**
*Here and elsewhere in the paper: I am not a big fan of these acronyms*
*WSMB, ASMB, SSMB. I feel it provides easier reading by just writing out 'winter SMB',*
*'annual SMB', and 'summer SMB'.*
**Authors' response:**
Done.

**Technical point #3 (page 3 - Table1):**
*Add some separation between the different stations, like a line, or a blank text line.*
**Authors' response:**
Done.

| Station | Location | Date of records | Timestep | Variables | Instrument | Manufacturer accuracy | Associated studies |
|---|---|---|---|---|---|---|---|
| AWS$_m$ | Moraine 2720 m a.s.l. | 2005-present | 30 min | Aspirated air T (°C)
 Relative humidity (%)
 Wind speed (m s$^{-1}$)
 and direction (degrees)
 Upward SW (W m$^{-2}$)
 Downward LW (W m$^{-2}$) | Vaisala HMP45C
 Vaisala HMP45C
 Young 05103
 Young 05103
 Kipp and Zonen CG3
 Kipp and Zonen CG3 | ±0.2°C
 3%
 0.3 m s$^{-1}$
 ±3°
 0.4%
 0.4% | *Six et al.* (2009)
 *Sicart et al.* (2008)
 *Dumont et al.* (2012) |
| AWS$_{g06}$ | Ablation area 2770 m a.s.l. | 9 July - 28 August 2006 | 30 min | Aspirated air T (°C)
 Relative humidity (%)
 Wind speed (m s$^{-1}$)
 and direction (degrees)
 Upward SW (W m$^{-2}$)
 Downward LW (W m$^{-2}$)
 EC measurements | Vaisala HMP45C
 Vaisala HMP45C
 Young 05103
 Young 05103
 Kipp and Zonen CG3
 Kipp and Zonen CG3
 Csat3 and Licor 7500 | ±0.2°C
 3%
 0.3 m s$^{-1}$
 ±3°
 0.4%
 0.4%
 0.4% | *Dumont et al.* (2012)
 *Litt et al.* (2016) |
| AWS$_{g-accu08}$ | Accumulation area 2900 m a.s.l. | 12 July - 10 September 2008 | 30 min | Aspirated air T (°C)
 Relative humidity (%)
 Wind speed (m s$^{-1}$)
 and direction (degrees)
 Upward SW (W m$^{-2}$)
 Downward LW(W m$^{-2}$) | Vaisala HMP45C
 Vaisala HMP45C
 Young 05103
 Young 05103
 Kipp and Zonen CG3
 Kipp and Zonen CG3 | ±0.2°C
 3%
 0.3 m s$^{-1}$
 ±3°
 0.4%
 0.4% | *Dumont et al.* (2012) |
| AWS$_{g08}$ | Ablation area 2770 m a.s.l. | 11 July - 2 August 2008 | 30 min | Aspirated air T (°C)
 Relative humidity (%)
 Wind speed (m s$^{-1}$)
 and direction (degrees)
 Upward SW (W m$^{-2}$)
 Downward LW (W m$^{-2}$) | Vaisala HMP45C
 Vaisala HMP45C
 Young 05103
 Young 05103
 Kipp and Zonen CG3
 Kipp and Zonen CG3 | ±0.2°C
 3%
 0.3 m s$^{-1}$
 ±3°
 0.4%
 0.4% | *Dumont et al.* (2012) |
| AWS$_{g09}$ | Ablation area 2770 m a.s.l. | 13 June - 4 September 2009 | 30 min | Aspirated air T (°C)
 Relative humidity (%)
 Wind speed (m s$^{-1}$)
 and direction (degrees)
 Upward SW (W m$^{-2}$)
 Downward LW (W m$^{-2}$) Albedo | Vaisala HMP45C
 Vaisala HMP45C
 Young 05103
 Young 05103
 Kipp and Zonen CG3
 Kipp and Zonen CG3 | ±0.2°C
 3%
 0.3 m s$^{-1}$
 ±3°
 0.4%
 0.4% | *Dumont et al.* (2012)
 *Litt et al.* (2016) |

**Technical point #4 (page 5 - l10):**
*Please rephrase this sentence on the instrumentation height adjustment. You probably mean that you lowered the instruments in order to keep the distance between instruments and the ice surface constant.*
**Authors' response:**
This part has been re-written in response to your comment and the comments of the other reviewer:
"Due to ice melt, instrument heights are not constant over time. However, at each station (except for AWS$_{g08-accu}$ where melt is limited), a sonic ranger was set up and helped determine the melt over each recorded time step. The heights of the instrument were then adjusted in our simulation using the melt

determined by the sonic ranger. Every 10 to 15 days, instruments were re-adjusted manually to a set height of 2 m."

**Technical point #5 (page 5  - l27):**
*'emitted long wave radiation and reflected short wave radiation', the earth surface does not emit short wave radiation.*
**Authors' response:**
The sentence has been re-written in response to your comment:
'…but the impact of emitted long wave radiation and reflected short wave radiation by surrounding slopes is not considered.'

**Technical point #6 (page 7 - l9):**
*could you replace 'explained' with 'caused'? 'explained' would require a more in-depth analysis*
**Authors' response:**
Done.

**Technical point #7 (page 9 - l24):**
*If I understand correctly, you have changed the lower albedo limit from 0.7 to 0.5, keeping the time decay. Then I suggest to replace 'fixed at' with 'set to', as 'fixed' could indicate that you eliminated the time evolution of the albedo.*
**Authors' response:**
You are right, 'fixed at' has been replaced by 'set to'.

**Technical point #8 (page 14 - Figure 4 caption):**
*Replace the hyphen in 'blue (a-c)', 'orange (b-d)', etc. with*
*a comma 'blue (a,c)'*
**Authors' response:**
In response to your comment and the remark made by the second reviewer, caption as been changed:
"**Figure 4.** Correlations between simulated (blue (a and c) for the ATI model and orange (b and d) for the Crocus model) and measured summer SMBs at each stake of Saint-Sorlin Glacier over the 2006-2015 period (a and b) and the 1996-2005 period (c and d). Circles represent measurements in the ablation area and solid dots represent measurements in the accumulation area."